# Equivariant Symmetry Breaking Sets

**YuQing Xie**                                                                    *xyuqing@mit.edu*
*Department of Electrical Engineering and Computer Science*
*Massachusetts Institute of Technology*

**Tess Smidt**                                                                    *tsmidt@mit.edu*
*Department of Electrical Engineering and Computer Science*
*Massachusetts Institute of Technology*

**Reviewed on OpenReview:** *https: // openreview. net/ forum? id= tHKH4DNSR5*

## Abstract

Equivariant neural networks (ENNs) have been shown to be extremely effective in applications involving underlying symmetries. By construction ENNs cannot produce lower symmetry outputs given a higher symmetry input. However, symmetry breaking occurs in many physical systems and we may obtain a less symmetric stable state from an initial highly symmetric one. Hence, it is imperative that we understand how to systematically break symmetry in ENNs. In this work, we propose a novel symmetry breaking framework that is fully equivariant and is the first which fully addresses spontaneous symmetry breaking. We emphasize that our approach is general and applicable to equivariance under any group. To achieve this, we introduce the idea of symmetry breaking sets (SBS). Rather than redesign existing networks, we design sets of symmetry breaking objects which we feed into our network based on the symmetry of our inputs and outputs. We show there is a natural way to define equivariance on these sets, which gives an additional constraint. Minimizing the size of these sets equates to data efficiency. We prove that minimizing these sets translates to a well studied group theory problem, and tabulate solutions to this problem for the point groups. Finally, we provide some examples of symmetry breaking to demonstrate how our approach works in practice. The code for these examples is available at https://github.com/atomicarchitects/equivariant-SBS.

## 1 Introduction

Equivariant neural networks have emerged as a promising class of models for domains with latent symmetry (Wang et al., 2022a). This is especially useful for scientific and geometric data, where the underlying symmetries are often well known. For example, the coordinates of a molecule may be different under rotations and translations, but the molecule remains the same. Traditional neural networks must learn this symmetry through data augmentation or other training schemes. In contrast, equivariant neural networks already incorporate these symmetries and can focus on the underlying physics. ENNs have achieved state-of-the-art results on numerous tasks including molecular dynamics, molecular generation, and protein folding (Batatia et al., 2022; Batzner et al., 2022; Daigavane et al., 2023; Ganea et al., 2021; Hoogeboom et al., 2022; Jia et al., 2020; Jumper et al., 2021; Liao & Smidt, 2022).

A consequence of the symmetries built-in to ENNs is that their outputs must have equal or higher symmetry than their inputs (Smidt et al., 2021). However, we frequently encounter situations where we desire a lower symmetry output given a higher symmetry input. We refer to an input-output pair where the output has lower symmetry as a symmetry breaking sample. Such symmetry breaking samples occur frequently in physical systems. In physics, these are classified into two types: explicit symmetry breaking and spontaneous symmetry breaking (Castellani et al., 2003). In explicit symmetry breaking, the governing laws are manifestly asymmetric explaining the symmetry breaking samples while in spontaneous symmetry breaking the laws

are symmetric but we still observe symmetry breaking samples. Notably, in spontaneous symmetry breaking there are multiple correct outputs for a given input and we have a modified form of equivariance. We expand on this in Section 2.

One class of approaches conducive for explicit symmetry breaking is learning to break symmetry. For example, Smidt et al. (2021) showed that the gradients of the loss function can be used to learn a symmetry breaking order parameter. This lets us identify what type of missing asymmetry is needed to correctly model the results. Another related approach is approximate and relaxed equivariant networks (Huang et al., 2023; van der Ouderaa et al., 2022; Wang et al., 2023; 2022b). These networks have similar architectures to equivariant models, but allow nontrivial weights in the layers to break equivariance. Hence, they can learn how much symmetry to preserve to fit the data distribution. However, since these methods explicitly break equivariance, they are not appropriate for spontaneous symmetry breaking. If all symmetrically related lower symmetry outputs are equally likely in the data, then relaxed networks will see the distribution as symmetric and fail to break symmetry. Further, since equivariance is broken, the method is not guaranteed to behave properly when shown data transformed under the group.

In the case of spontaneous symmetry breaking, there exist some works which partially solve the problem. Balachandar et al. (2022) design a symmetry detection algorithm and an orientation aware linear layer for mirror plane symmetries. However, the scope of their methods are specific to mirror symmetries and point cloud data. Finally, Kaba & Ravanbakhsh (2023) take the approach of defining a relaxed version of equivariance which takes input symmetry into account. They derive modified constraints linear layers would need to satisfy for this relaxed equivariance and argue such models can give lower symmetry outputs. However, they mention these conditions do not reduce as easily as for the usual equivariant linear layers. Further this method still does not provide a mechanism to sample all possible outputs.

Our work focuses on the spontaneous symmetry breaking case. This case is extremely important as spontaneous symmetry breaking occurs in many physical phenomena (Beekman et al., 2019). Hence the widespread adoption of machine learning techniques for scientific applications will inevitably run into symmetry breaking issues. Examples of existing applications which may run into difficulties include crystal distortion, predicting ground state solutions from Hamiltonians, and solving PDEs (Jafary-Zadeh et al., 2019; Lewis et al., 2024; Lino et al., 2022). In the crystal distortion case, there are distortions from high symmetry to low symmetry structures (Kay & Vousden, 1949). The lowest energy states of physical systems (ground states of the Hamiltonian) are often low symmetry and famously explains the Higgs mechanism for giving particles mass (Higgs, 1964). Finally, Karman vortex sheets are a well known example of symmetry breaking in fluid simulations (Tang & Aubry, 1997).

In this work, we provide the first general solution for the spontaneous symmetry breaking problem in equivariant networks. We identify that the key difficulty is how to allow equivariant networks to output a set of valid lower symmetry outputs. Similar to Smidt et al. (2021), we use the natural idea of providing additional symmetry breaking parameters as input to the model. However, rather than learning these parameters, we show that we can sample them from a symmetry breaking set (SBS) that we design based only on the input and output symmetries. In particular, we prove that minimizing the size of the equivariant SBSs is equivalent to a fundamental group theory question. We emphasize that this fully characterizes how to efficiently break symmetry with equivariant SBSs for any group. Counter-intuitively, we find that it is sometimes beneficial to break more symmetry than needed.

The main features of our method are the following:

1. **Equivariance:** Our framework is fully equivariant. That we can achieve this is a key point of this work and allows simulation of spontaneous symmetry breaking.

2. **Simple to implement:** Our approach only requires a designing a set of additional inputs into an equivariant network. We have fully characterized the design of such sets.

3. **Generalizability:** We emphasize that our characterization of SBSs applies to any groups.

We would like to point out that to achieve our results, we assume we can detect the symmetry of our input and outputs. Further, there is the more general problem of treating symmetrically related outputs as the same in our loss. We discuss these limitations in Appendix D.

The rest of this paper is organized as follows. In Section 2 we formalize the symmetry breaking problem, the distinction between explicit and spontaneous symmetry breaking, and the type of task performed in the spontaneous symmetry breaking setting. In Section 3, we examine the case where we break all symmetries of our input. We motivate the idea of a SBS and show that imposing equivariance leads to an additional constraint of closure under the normalizer. The intuition is that the normalizer characterizes all orientations of our data which do not change its symmetry. Next, we translate bounds on the size of the equivariant SBSs into the purely group theoretical problem of finding complements. We have tabulated these complements in Appendix F for the point groups. In Section 4, we generalize to the case where we may still share some symmetries with our input. In Section 5 we describe how to construct SBSs in an actual implementation. Having fully described our framework, we then highlight how our method relates to prior symmetry breaking works in Section 6. Finally, in Section 7, we introduce examples of symmetry breaking and demonstrate how our method works in practice.

**Notation and background:** An overview of the notation and common symbols used can be found in Appendix A. A brief overview of mathematical concepts needed in the paper can be found in Appendix B and an overview of ENNs can be found in Appendix C.

## 2 Symmetry breaking problem

Here, we make precise what we mean by a symmetry breaking and the issue it poses for equivariant neural networks. We begin with the following observation first made in Smidt et al. (2021).

**Lemma 2.1.** *Let $X$ and $Y$ be spaces equipped with a group action of $G$. We can choose an equivariant $f : X \to Y$ such that $f(x) = y$ only if $\operatorname{Stab}_G(y) \geq \operatorname{Stab}_G(x)$.*

See Appendix E.1 for a generalization of this lemma and a proof.

Hence, the output of an equivariant function must have at least the symmetry of the input. This motivates the following definition of symmetry breaking at the individual sample level.

**Definition 2.2** (Symmetry breaking sample)**.** Let $G$ be a group. A sample with input $x$ and output $y$ is symmetry breaking with respect to $G$ if $\operatorname{Stab}_G(y) \not\geq \operatorname{Stab}_G(x)$.

Lemma 2.1 tells us a symmetry breaking sample with respect to $G$ can never be perfectly modeled by a $G$-equivariant function. In experimental samples, we may have random noise in our observations of our outputs which causes samples to be symmetry breaking. In such cases equivariance is beneficial since it can help remove the noise. However, in some cases we truly have a symmetry breaking sample even with a perfect measurement. In physics this is classified into two cases: explicit symmetry breaking and spontaneous symmetry breaking. In the following discussion, it is useful to view the underlying model as a set valued function. A typical function $f : X \to Y$ can be thought of as a set valued function $F : X \to \mathcal{P}(Y)$ defined as $F(x) = \{f(x)\}$. Note that if there is an action of $G$ on $Y$, one can naturally define an action on $\mathcal{P}(Y)$ such that $U \subseteq Y$ transforms as $gU = \{gu : u \in U\}$. Hence there is a natural way to define equivariance of set valued functions.

In explicit symmetry breaking, the underlying physics of the system is asymmetric. For example, there may be an unknown electric field which breaks rotation symmetry.

**Definition 2.3** (Explicit symmetry breaking)**.** Let $G$ be a group. A function $F : X \to \mathcal{P}(Y)$ which is not $G$-equivariant explicitly symmetry breaks $G$.

In such cases using an equivariant function actually prevents us from learning the true function.

In spontaneous symmetry breaking, the underlying physics of the system is symmetric, however there is a set of stable lower symmetry outputs which are equally likely.

**Definition 2.4** (Spontaneous symmetry breaking (SSB) function)**.** Let $G$ be a group with actions defined on spaces $X, Y$. Let $F : X \to \mathcal{P}(Y)$ be $G$-equivariant. We say $F$ spontaneous symmetry breaks at $x$ if there is some $y \in F(x)$ such that $\text{Stab}_G(x) > \text{Stab}_G(y)$.

The key things here are that the set valued function $F$ is equivariant even though individual observed samples $(x, y)$ for $y \in F(x)$ may be symmetry breaking. Hence, our problem becomes the following.

**Problem:** How does one create an equivariant architecture which can output a set of possibly lower symmetry outputs?

## 2.1 Examples of explicit and spontaneous symmetry breaking

### 2.1.1 Double well potential

The double well potential is a classic example of symmetry breaking in physics. Consider a 1-dimensional energy potential defined as $U(x) = x^4 - 2x^2 + 1$. Note that this system has reflection symmetry about $x = 0$ since substituting $x \to -x$ does not change the potential

$$U(-x) = (-x)^4 - 2(-x)^2 + 1 = x^4 - 2x^2 + 1 = U(x).$$

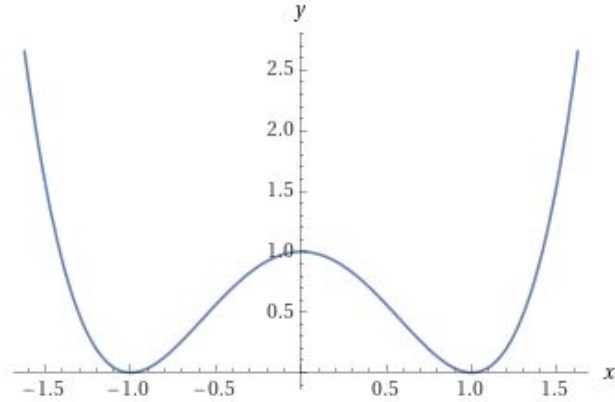

Figure 1: A classic double well potential of $x^4 - 2x^2 + 1$. Clearly this potential has reflection symmetry. The two minima are $x = 1$ and $x = -1$.

In physics we often care about finding the ground state solutions which corresponds to the value of $x$ minimizing any given potential. In the potential above, we see that

$$U(x) = x^4 - 2x^2 + 1 = (x^2 - 1)^2 = (x - 1)^2(x + 1)^2$$

so there two global minima at $x = \pm 1$. However, individually these minima break reflection symmetry since $1 \neq -1$. Hence if our sample consists of input potential $U(x)$ (perhaps encoded using polynomial coefficients) and output $x = 1$, then it is a symmetry breaking sample with respect to reflection. Similarly the input-output pair of $U(x), x = -1$ would also be a symmetry breaking sample.

Note that by Lemma 2.1, a network equivariant to reflection across $x = 0$ would be unable to directly output the minima. Such a network realizes that $1$ and $-1$ are symmetric and would return the average of $0$.

Perhaps in some cases, we decide to only consider positive solutions. In this case we explicitly break the reflection symmetry and methods such as finding order parameters and relaxed equivariance learn to break such symmetry Huang et al. (2023); van der Ouderaa et al. (2022); Smidt et al. (2021); Wang et al. (2022b; 2023). Since these learned functions are not equivariant, they are explicitly symmetry breaking by Definition 2.3.

However, in many other cases both solutions are valid. In this case, the set $\{1, -1\}$ of minima is actually invariant under reflection but the individual solutions we want are not. We would satisfy Definition 2.4 since

for a set-valued function returning both minima we remain equivariant, while picking either of the individual minima gives a symmetry breaking sample.

### 2.1.2 Ferromagnetism

A good physical example of symmetry breaking is ferromagnetism. Individual atoms in such materials have a magnetic dipole moment and due to strong interaction between neighboring atoms, the moments tend to align at cool enough temperatures Aharoni (2000). Hence, if we heat and cool a ferromagnetic material, we obtain regions where the dipole moments of individual atoms align. These regions are known as magnetic domains. Because the dipoles are aligned, there is a net magnetic moment in the domain which breaks rotational symmetry.

In the presence of a strong external magnetic field **B**, the moments of the magnetic domains tend to align with the external field. This is an example of explicit symmetry breaking since the external field explicitly breaks rotational symmetry. From the perspective of Definition 2.3 we have a non-equivariant function since we prefer the direction which aligns with the external field.

However, if there is no external magnetic field, the direction of the magnetic moment in each domain is uniformly random. This is an example of spontaneous symmetry breaking, the governing laws are symmetric yet we observe asymmetry in the individual domains. From the perspective of Definition 2.4 we have a set of vectors in all orientations as valid moments and we do not need to break equivariance. However choosing a particular moment would give a symmetry breaking sample.

These two cases are depicted in Figure 2.

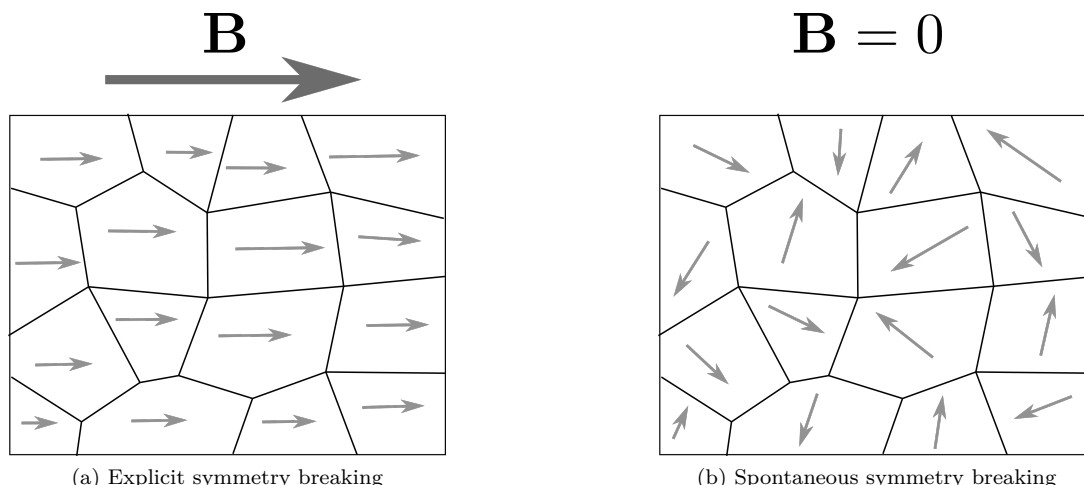

(a) Explicit symmetry breaking                              (b) Spontaneous symmetry breaking

Figure 2: (a) Example of explicit symmetry breaking. Magnetic moment in domains align with a strong external magnetic field **B**. The external field explicitly breaks symmetry of the system. (b) Example of spontaneous symmetry breaking. Presence of a moment in each domain breaks rotational symmetry. However, there is no magnetic field so governing laws of the system are symmetric. Consequently the observed moments are uniformly random in orientation.

## 3 Fully broken symmetry

First, we consider the case where we break all symmetry of our input. Here, our desired outputs $y$ share no symmetry with $x$. In other words $\mathrm{Stab}_S(y) = \{e\}$. This will lay the foundation for analyzing the general case of partially broken symmetry. Let the symmetry group of our data $x$ be $S$.

### 3.1 Symmetry breaking set (SBS)

When there is symmetry breaking there are multiple equally valid symmetrically related outputs. The purpose of a symmetry breaking object is to allow an equivariant network to pick one of them. In principle, we want all symmetrically related outputs to be equally likely so it makes sense to think of a set $B$ of symmetry breaking objects we sample from.

For any $s \in S$ and $b \in B$, since $sb$ is symmetrically related to $b$ it is natural to also include it in $B$. Hence, acting with $s$ on the elements of $B$ should leave the set unchanged. Further, for any $b \in B$, the stabilizer $\text{Stab}_S(b)$ must be trivial since we want to break all symmetries of our input. This is exactly the definition of a free group action of $S$ on $B$. Hence, we define a symmetry breaking set as follows.

**Definition 3.1** (Symmetry breaking set)**.** Let $S$ be a symmetry group. Let $B$ be a set of elements which $S$ acts on. Then $B$ is a symmetry breaking set (SBS) if the action of $S$ on $B$ is a free action.

### 3.2 Equivariant SBS

However, the above definition of an SBS is insufficient when considering equivariance. Here, we show that we need the stronger constraint of closure under the normalizer.

To illustrate the problem, consider a network which is $SO(3)$ equivariant and a triangular prism aligned so that the triangular faces are parallel to the $xy$ plane. Suppose our task was just to pick a point of the prism. A naive way to break the symmetry is to have an ordered pair of unit vectors. The first vector is in the $xy$ plane and points towards one of the triangle vertices. The second vector points up or down in the $z$ direction, corresponding to the upper or lower triangle.

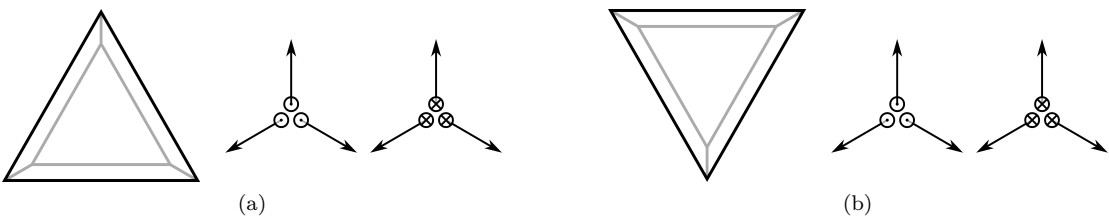

(a)           (b)

Figure 3: (a) Naive way to break symmetry in a triangular prism where one vector points to a vertex of a triangle and a second vector points to the lower or outer triangle. (b) A rotated version of the triangular prism in. Note that the same symmetry breaking objects now point to edges of the triangle rather than vertices. However, both prisms have the exact same symmetry elements.

However, consider the same prism but rotated 180° around $z$. We can check that the symmetry groups are exactly the same so we want the same SBS. But the symmetry breaking objects are related differently. In the second prism, the first vector points to an edge rather than a vertex. For equivariance, our symmetry breaking objects should be related to both prisms in the same way. So our choice of SBS was not equivariant.

Here, one may simply choose a canonical orientation and decide that we will rotate the original SBS by 180° in the latter case. However, our input may be arbitrarily complicated, and it may be hard to decide on a canonicalization. Further, canonicalization may introduce discontinuities. Hence, we would like to construct SBSs to be only dependent on the symmetry of our data, not how our data is represented. To understand exactly what additional condition is necessary, we need to carefully investigate how the symmetry breaking scheme works.

Let $f$ be our $G$-equivariant function and $x$ be our input data. Suppose we know the symmetry $S$ of our input. Let $\mathbf{B}$ be some set with a group action of $G$ defined on it. We would like to obtain our set of symmetry breaking objects based on just information about the input symmetry. So suppose we have a function $\sigma : \text{Sub}(G) \to \mathcal{P}(\mathbf{B})$ that does so. This function takes in a subgroup symmetry and gives a SBS composed of elements from $\mathbf{B}$. Then the symmetry breaking step happens when we take a random sample $b$

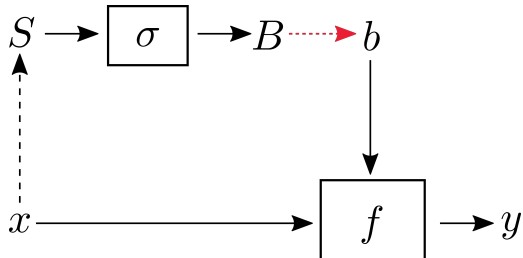

Figure 4: Diagram of how we might structure our symmetry breaking scheme. From our data $x$, we may obtain its symmetry $S$. This $S$ is then fed into a function $\sigma$ which gives us the set of symmetry breaking objects needed. We sample a $b$ from this set breaking the symmetry of our input and feed this $b$ along with the input $x$ into our equivariant function $f$. Finally we obtain an output $y$ which has lower symmetry than the input $x$.

from the SBS. This symmetry breaking object is then fed into our equivariant function, allowing it to break symmetry. A diagram of this process is shown in Figure 4.

Certainly, since we break the symmetry of our input data, we break equivariance. However, imagine we give our function all possible symmetry breaking objects and collect at the end the set $Y$ of all outputs our model gives. This process shown in Figure 5 would then not need to break any symmetry. This is because all outputs as a set has the same symmetry as the input. Hence we can impose equivariance on our process.

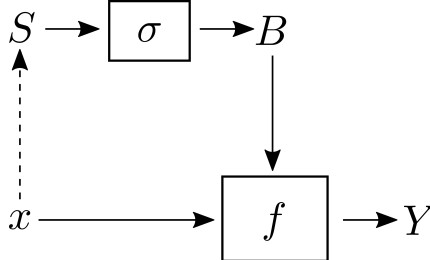

Figure 5: Diagram of how we break symmetry, but now we keep all possible outputs.

It is well known that the composition of equivariant functions remains equivariant. Hence, we just need to impose equivariance on $\sigma$. In order to do so, we must understand how the input and output transform. Suppose we act on our data with some group element $g \in G$. Then it becomes $gx$. Since $S$ is the symmetry of our original data, we find $gx = gsx = (gsg^{-1})(gx)$ for any $s \in S$. So the symmetry of the transformed data is $gSg^{-1}$. Hence, the input of $\sigma$ transforms as conjugation. Next, recall the output of $\sigma$ is some subset $B$ of elements of $\mathbf{B}$. Since there is a group action for $G$ defined on $\mathbf{B}$, we can define an action on $B$ by just acting on its elements and forming a new set. Figure 6 shows how our procedure would change if it were equivariant, and we act on our input by some group element $g$.

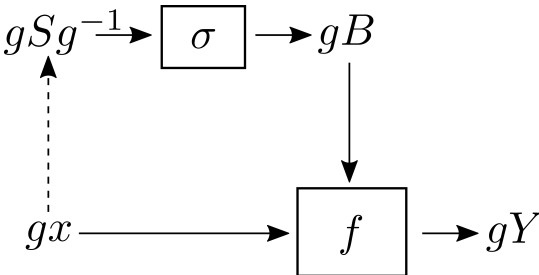

Figure 6: Diagram of what happens when we act on the input with some group element $g$.

We can now clearly see the issue. The input into $\sigma$ transforms under conjugation, and the stabilizer of a subgroup under conjugation is precisely the definition of the normalizer $N_G(S)$. However, in many cases $N_G(S)$ is a supergroup of $S$. Therefore by Lemma E.1, our SBS not only needs to be invariant under $S$, but also be invariant under $N_G(S)$. See Appendix E.2 for a more formal justification.

Hence, we will call a SBS equivariant if it can be the output of an equivariant $\sigma$. The intuition above tells us this amounts to closure under the normalizer $N_G(S)$.

**Definition 3.2** (Equivariant symmetry breaking sets)**.** Let $S$ be a subgroup symmetry of a group $G$ and $\mathbf{B}$ be a set with an action of $G$ defined on it. Let $B \subset \mathbf{B}$ be a SBS. Then $B$ is $G$-equivariant if $\forall g \in N_G(S)$ we have $B = gB$.

### 3.3 Ideal case and complement of normal subgroups

Now that we know how to equivariantly break a symmetry, we would like to understand how to do so efficiently. Intuitively, we expect a smaller SBS to be better. If we have a larger SBS, multiple symmetry breaking objects map to the same output so the network needs to learn that these are the same. Reducing the SBS would decrease the equivalences our network needs to learn. In the ideal case, exactly one symmetry breaking parameter corresponds to each output. Since our outputs are related by symmetry transformations (transitive) under $S$, this corresponds to the equivariant SBS being transitive under $S$. It turns out, we can equate constructing ideal equivariant SBSs to the constructing complements of normal subgroups. A slightly weaker version of this statement can be found in Theorem 3.1.4 of Kurzweil & Stellmacher (2004).

**Theorem 3.3.** *Let $G$ be a group and $S$ a subgroup. Let $B$ be a $G$-equivariant SBS for $S$. Then it is possible to choose an ideal $B$ if and only if $S$ has a complement in $N_G(S)$.*

*If $b$ is an element where $\mathrm{Stab}_{N_G(S)}(b)$ is a complement of $S$ in $N_G(S)$, then $\mathrm{Orb}_S(b)$ is an ideal $G$-equivariant SBS.*

*Remark* 3.4. It turns out the complement if it exists is isomorphic to $N_G(S)/S$. We can intuitively think of $N_G(S)/S$ as giving all possible orientations of our data such that its symmetry remains unchanged.

The proof of this theorem is in Appendix E.3. Finding complements of normal subgroups is a well studied group theory problem Kurzweil & Stellmacher (2004). For the point groups, which are the finite subgroups of $O(3)$, we have tabulated the complements if they exist in Appendix F.

The intuition for this theorem comes from the following observation. Essentially equivariant SBSs consist of orbits of $N_G(S)$. By the orbit-stabilizer theorem, we can reduce the size of an orbit by making an element $b$ generating the orbit more symmetric. However, $b$ must still fully break the symmetry of $S$. The complement, if it exists, is essentially the maximal symmetry $b$ can have while still breaking the symmetry of $S$.

### 3.4 Nonideal equivariant SBSs

In the case where we cannot achieve an ideal equivariant SBS, we would still like to characterize how efficient it is. To do this, we define what we call the degeneracy of an equivariant SBS. In general, each orbit under $S$ gives us one SBS which can be matched one to one to our outputs.

**Definition 3.5** (Degeneracy)**.** Let $B$ be a $G$-equivariant SBS for $S$. We define the degeneracy to be

$$\mathrm{Deg}_S(B) = |B/S|.$$

Note that an ideal equivariant SBS $B_{ideal}$ (if it exists) has exactly 1 orbit of $S$, so $\mathrm{Deg}_S(B_{ideal}) = 1$. We would also like to understand how small we can make the degeneracy if we cannot make it 1. It turns out Theorem 3.3 allows us to convert this to a group theory problem.

**Corollary 3.6.** *Let $G$ be a group and $S$ a subgroup. Let $M$ be such that $S \leq M \leq N_G(S)$. Let $B$ be a $G$-equivariant SBS for $S$ which is transitive under $N_G(S)$. Then it is possible to choose $B$ such that every $S$-orbit is also a $M$-orbit if and only if $S$ has a complement in $M$. In particular,*

$$\mathrm{Deg}_S(B) \leq |N_G(S)/M|.$$

See Appendix E.4 for a proof. In the ideal case we can make $M$ to be $N_G(S)$ so the above formula gives an degeneracy of 1 as expected.

## 4 Partially broken symmetry

We can now use our framework for full symmetry breaking to understand the case of partial symmetry breaking. In this case, our desired output may share some nontrivial subgroup symmetry $K \leq S$ with our input. Note the case of $K = \mathbf{1}$ corresponds to full symmetry breaking and $K = S$ corresponds to no symmetry breaking.

### 4.1 Partial SBS

Similar to the full symmetry breaking case, we would like to create a set of objects which we can use to break our symmetry. Now we can relax the restriction of free action. Intuitively, we can allow our symmetry breaking objects to share symmetry with our input, as long as it is lower symmetry than our outputs. However, the symmetrically related outputs may be invariant under different subgroups of $S$. Recall that if some element $y$ gets transformed to $sy$, its stabilizer $K$ gets transformed to $sKs^{-1}$. Hence, the stabilizers of the outputs are the subgroups conjugate to $K$ under $S$ denoted as $\mathrm{Cl}_S(K)$. Based on this intuition, we can define partial SBS as follows.

**Definition 4.1** (Partial SBSs). Let $S$ be a symmetry group and $K$ a subgroup of $S$. Let $P$ be a set of elements with an action by $S$. Then $P$ is a $K$-partial SBS if for any $p \in P$, there exists some $K' \in \mathrm{Cl}_S(K)$ such that $K' \geq \mathrm{Stab}_S(p)$.

Certainly, a full SBS is a partial one as well since the stabilizers of all its elements under $S$ is the trivial group. In general, we can always break more symmetry than needed and still obtain our desired output. However, it is useful to consider the case where we only break the necessary symmetries. Counter-intuitively, we discuss in Section 4.5 that this turns out to not always be optimal.

**Definition 4.2** (Exact partial SBS). Let $S$ be a group and $K$ a subgroup of $S$. Let $P$ be a $K$-partial SBS for $S$. We say $P$ is exact if for all $p \in P$, we have $\mathrm{Stab}_S(p) \in \mathrm{Cl}_S(K)$.

### 4.2 Equivariant partial SBS

Similar to before, we define equivariant partial SBSs. The idea is the same, but now we need to identify the symmetry of the input and the set of conjugate symmetries for the output. Define

$$\mathrm{SubCl}(G) = \{(S, \mathrm{Cl}_S(K)) : S \in \mathrm{Sub}(G), K \leq S\}.$$

Let $\mathbf{P}$ be a set with a group action of $G$ defined on it. As before, the idea is that we have an function $\pi : \mathrm{SubCl}(G) \to \mathcal{P}(\mathbf{P})$ which outputs our partial SBS. The condition of equivariance for our partial SBS is imposing equivariance on $\pi$.

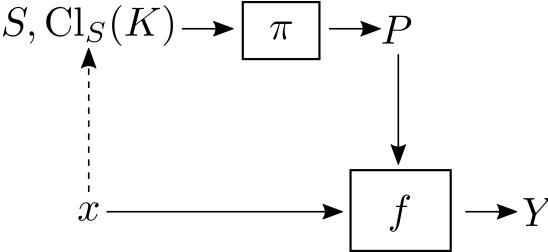

Figure 7: Diagram of how we perform partial symmetry breaking. Here, we need to specify not just the symmetry of our input but also the symmetries of our output. Since any of our outputs are equally valid, it only makes sense to specify the set of conjugate subgroups $\mathrm{Cl}_S(K)$ our outputs are symmetric under.

The symmetry breaking scheme is depicted in Figure 7. As before, we can impose equivariance on this diagram. We need to know how $\mathrm{Cl}_S(K)$ transforms. Note that if our input gets acted by $g$, we expect the outputs to also get acted by $g$. Since $K$ is the stabilizer of one of the outputs, we expect $K$ to transform to $gKg^{-1}$. Hence we have the transformation

$$\mathrm{Cl}_S(K) \to \mathrm{Cl}_{gSg^{-1}}(gKg^{-1}).$$

Similar to before, by Lemma E.1 we need the output of $\pi$ to also be invariant under the stabilizer of the input. Noting that the normalizer is defined as the stabilizer of $S$ under conjugation, we can define a generalized normalizer as the stabilizer of $S, \mathrm{Cl}_S(K)$.

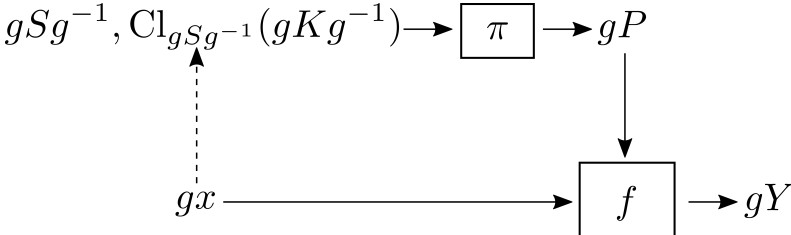

Figure 8: Diagram of how our symmetry scheme changes when we transform our input by some group element $g \in G$.

**Definition 4.3** (Generalized normalizer). Define the generalized normalizer $N_G(S, K)$ to be

$$N_G(S, K) = \{g : gKg^{-1} \in \mathrm{Cl}_S(K), g \in N_G(S)\}.$$

We can now define equivariant partial SBSs using closure under this generalized normalizer. See Appendix E.5 for a more formal justification.

**Definition 4.4** (Equivariant partial SBSs). Let $S$ be a subgroup symmetry of a group $G$. Let $P$ be a $K$-partial SBS. Then $P$ breaks the symmetry $G$-equivariantly if $\forall g \in N_G(S, K)$ we have $P = gP$.

Note that closure under $N_G(S, K)$ is a weaker condition than closure under $N_G(S)$. Hence any equivariant full SBS is also an equivariant $K$-partial SBS for any $K$.

### 4.3 Ideal equivariant partial SBS

Similar to the full symmetry breaking case, we ideally would like to have a one to one correspondence between elements in our equivariant SBS and our symmetrically related outputs. For this to happen, we clearly need our SBS to be exact and for our SBS to be transitive under $S$. We can generalize Theorem 3.3 to obtain a necessary and sufficient condition to have an ideal equivariant partial SBS.

**Theorem 4.5.** *Let $G$ be a group and $S$ and $K$ be subgroups $K \leq S \leq G$. Let $P$ be a $G$-equivariant $K$-partial SBS. Then we can choose an ideal $P$ (exact and transitive under $S$) if and only if $N_S(K)/K$ has a complement in $N_{N_G(S,K)}(K)/K$.*

*If $p$ is an element such that $\mathrm{Stab}_{N_G(S,K)}(p)/K$ is a complement of $N_S(K)/K$ in $N_{N_G(S,K)}(K)/K$, then $\mathrm{Orb}_S(p)$ is an ideal $G$-equivariant $K$-partial SBS*

See Appendix E.6 for a proof.

### 4.4 Nonideal equivariant partial SBS

Similar to the full symmetry breaking case, when we cannot achieve an ideal equivariant partial SBS we want to characterize how efficient our nonideal partial SBS is. Again, the idea is that in the nonideal case, our network needs to map multiple symmetry breaking objects to the same output. We define the degeneracy of $P$ to quantify this multiplicity.

**Definition 4.6** (Degeneracy)**.** Let $G$ be a group, $S$ be a subgroup, and $K$ a subgroup of $S$. Let $P$ be a $G$-equivariant $K$-partial SBS for $S$. Let $T$ be a transversal of $S/K$. Let $P_t$ be such that every $p \in P$ is uniquely written as $p = tp_t$ for some $t \in T$ and $p_t \in P_t$. Then we define

$$\mathrm{Deg}_{S,K}(P) = |P_t|.$$

The intuition for this definition is that $P_t$ is the set of objects which together with our input may get mapped to some output $y$ by our equivariant network. In other words, we have $f(x, P_t) = \{y\}$ for equivariant $f$ and all other $P$ get mapped to different symmetrically related outputs. Without loss of generality assume $y$ has $\mathrm{Stab}_S(y) = K$. Then for any symmetrically related output $ty$ (where $t \in T$), we can see from equivariance of $f$ that $f(x, tP_t) = \{ty\}$. It is now clear that the size of $P_t$ counts how many symmetry breaking objects must be mapped to the same output.

Note that in the case $K = \mathbf{1}$, $S/K = S$ so $P_t$ just consists of representatives from $P/S$. So this reduces to the degeneracy defined for full SBS. Also, note that in the ideal case, there is exactly one symmetry breaking object for each output. So degeneracy is 1 in that case.

We would like to derive bounds on the degeneracy of our equivariant partial SBSs. Similar to the full SBS case, we use Theorem 4.5 to convert this into a group theory question.

**Corollary 4.7.** *Let $G$ be a group, $S$ a subgroup, and $K$ a subgroup of $S$. Let $K'$ be a subgroup of $K$ and $M$ a subgroup of $N_G(S, K) \cap N_G(S, K')$ which contains $S$. Suppose $P$ is a $G$-equivariant $K$-partial SBS for $S$ which is transitive under $N_G(S, K)$. We can choose $P$ such that $\mathrm{Stab}_S(p) \in \mathrm{Cl}_{N_G(S,K)}(K')$ for all $p$ and all $S$-orbits in $P$ are also $M$-orbits if and only if $N_S(K')/K'$ has a complement in $N_M(K')/K'$. Further, such a $P$ has*

$$\mathrm{Deg}_{S,K}(P) \leq |K/K'| \cdot |N_G(S, K)/M|.$$

See Appendix E.7 for a proof.

### 4.5 Optimality of exact partial symmetry breaking

Note that in the previous section, we have been very careful to allow our partial SBS to break more symmetry than needed. Intuitively, we would like to say that it is always optimal to break down exactly to the symmetry of our output. That is, we only need to consider exact partial SBSs.

Certainly, ignoring any equivariance constraints, given any non-exact $K$-partial SBS, we can construct an exact $K$-partial SBS by picking an element $b$ with $\mathrm{Stab}_S(b) \leq K$ and identifying its orbit under $K$ together as one partial symmetry breaking object $p = Kb$. We construct the orbit of $p$ under action by $S$ as our $K$-partial SBS.

We might expect that some modification of this construction can convert any non-exact equivariant $K$-partial SBS into an exact equivariant $K$-partial symmetry breaking one. Naively, we just take the orbit of the elements in the construction above under $N_G(S, K)$ to obtain $G$-equivariance. However, in Appendix G we come up with an explicit example where no exact equivariant $K$-partial SBS is smaller than the best equivariant full SBS.

## 5 Constructing SBSs

Now that we have fully characterized what equivariant symmetry breaking sets are, we show how to construct them. We focus on $G = O(3)$ in this section though the ideas here are applicable for other groups as well.

### 5.1 Expressing subgroups

First, it is important to have a way of expressing subgroup of $G$, the group we want to be equivariant under. The subgroups of $O(3)$ are well studied and have been completely classified Hahn et al. (1983). In particular,

there are 7 infinite axial families of finite point groups and 7 additional finite ones. There are only 5 infinite subgroups which are closed. However, the names of these subgroups do not specify how they are "oriented" in $O(3)$. Hence, we propose to represent the subgroups in the following way. We first choose a canonical orientation of the classified point groups.

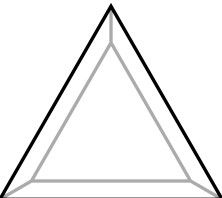 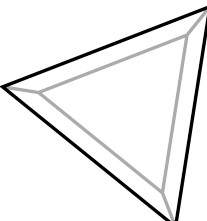

Figure 9: Two identical triangular prisms differing by a rotation. Both have symmetry $D_{6h}$ by name, however the actual symmetry axes differ.

A standard choice is inspired by the Hermann-Mauguin naming scheme for point groups. For the 7 infinite series of axial groups, we choose to align the high order symmetry axis along the $z$-axis. If in addition there are 2-fold rotations, we choose one of them to be along the $x$-axis. If there are no 2-fold rotations but there are mirror plane parallel to the $z$-axis, we choose the $yz$-plane to be in the group. Of the remaining 7 point groups, 5 are cubic groups. For these we can choose the cube they leave invariant to have sides perpendicular to one of the $x, y, z$ axes. Finally, the remaining 2 point groups are the icosahedral groups with and without inversion. For these we can choose to align a 5-fold axis with the $z$-axis and a 3-fold axis with the $x$-axis.

Next, for any point group $S$ with arbitrary alignment, there is always some $g \in O(3)$ (in fact we can always pick $g \in SO(3)$) such that $g^{-1}Sg$ brings it to the canonical orientations defined above. Hence, we can always express an arbitrary point group $S$ as a pair $g, \texttt{name}(S)$ of a rotation and the name of the group.

To generalize to arbitrary groups, we note that the point groups classify the classes of conjugate subgroups $\text{Cl}_{O(3)}(S)$ of $O(3)$. Hence for general $G$, we need to classify the corresponding classes of conjugate subgroups $\text{Cl}_G(S)$ and choose a canonical representative for each class in order to apply scheme outlined in this section.

## 5.2 Representing a set of conjugate subgroups

In the partial symnmetry breaking case, we also provide a set of conjugate subgroups. Similar to our notation $\text{Cl}_S(K)$, we can specify a set of conjugate subgroups with $(S, K)$, where subgroups $S$ and $K$ are represented in the way described previously.

## 5.3 Representing a SBS

The idea is very simple. We start with some object which breaks enough symmetry for our task. To satisfy the equivariance condition of closure under the normalizer or generalized normalized, we can simply take the orbit as our SBS or partial SBS. In principle a SBS can consist of multiple such orbits, but we can always only use one orbit as a SBS and multiple orbits increase the degeneracy. Hence we assume all our SBSs consist of one orbit and we can fully specify a SBS as a pair $(b, N)$ where $b$ is a symmetry breaking object and $N$ is a group over which we take the orbit of $b$.

In the case of finite group $N$, we can explicitly compute the elements in the orbit. However, if $N$ is infinite then this does not work. In practice, it is usually enough that we can sample lower symmetry outputs. Hence, it suffices to be able to sample the SBS which we can do by sampling an element from $N$.

## 5.4 Constructing an equivariant full SBS

We would like to construct a full SBS given an input symmetry $S = (g, \texttt{name}(S))$. However, we want to do so in an equivariant way. One way to achieve this is to first consider only any input group in its canonical orientation and construct a SBS $B$ for it. Then simply returning $gB$ would guarantee that our construction

is equivariant. Hence, we just need to construct a canonical SBS for each possible point group. Note that this can be viewed as a case of equivariance by canonicalization Kaba et al. (2023).

Note an equivariant full SBS needs closure under a normalizer. If we work with $O(3)$ equivariance, we simply look up the normalizer $N_{O(3)}(S)$ from Table 4. If we are equivariant under some subgroup $G \subset O(3)$ then we note that $N_G(S) = N_{O(3)}(S) \cap G$ which can be used to compute the desired normalizer. All that remains is how to specify a canonical symmetry breaking object for each point group in canonical orientation. If we have such an object, then we obtain the following algorithm for creating equivariant full SBSs. Let `Normalizers` be a function which takes in a name of a point group and gives the corresponding normalizer classified in Table 4.

---

**Algorithm 1** Equivariant full SBS

---

**Input**
  $S$          Symmetry of input expressed as pair $(g, \mathtt{name}(S))$
  $b$          Canonical symmetry breaking object
**Output**
  $B$          Symmetry breaking set expressed as a pair $(b', N)$
$(n, \mathtt{name}(N_{O(3)}(S))) \leftarrow \mathtt{Normalizers}[\mathtt{name}(S)]$
$N \leftarrow (gn, \mathtt{name}(N_{O(3)}(S)))$
**return** $(gb, N)$

---

In general, the choice of a canonical symmetry breaking object is flexible. To satisfy the definition, one just needs the corresponding object for $\mathtt{name}(S)$ to not share any symmetries with $S = (e, \mathtt{name}(S))$. However, as discussed in Section 3.3, an ideal SBS should be more efficient than a nonideal one. Hence, if possible we would like to pick such an object so that it generates an ideal SBS. Theorem 3.3 tells us exactly the conditions needed to choose an ideal SBS. In particular, we would need the additional condition that $b$ have the symmetry of a complement $H$ while not having the symmetry of $S$. We fully characterized the relevant cases in Appendix F.

### 5.5 Constructing equivariant partial SBS

In the partial symmetry breaking case, we want to obtain a partial SBS from the symmetry of the input and the set of conjugate subgroup symmetries of the outputs. Importantly, note that we want closure under $N_G(S, K)$ rather than under $N_G(S)$. To compute $N_G(S, K)$, we use the following fact.

**Lemma 5.1.** *We have the following formula*

$$N_G(S, K) = S(N_G(S) \cap N_G(K)).$$

See Appendix E.8 for a proof.

Thus we get the following algorithm for computing an equivariant partial SBS.

We emphasize that for any $K' < K$, a $K'$-partial SBS can also serve as a $K$-partial SBS since we can always break more symmetry than needed. In particular, a full SBS often suffices for simplicity.

Similar to the full SBS case, we have flexibility in choosing our canonical object used to generate the partial SBS. To satisfy the definition, all we need is for $\mathrm{Stab}_G(p) \leq K'$ for some $K' \in \mathrm{Cl}_S(K)$. An ideal partial SBS is desirable, especially if we wish to ensure we do not break any extra symmetry. The condition given by Theorem 4.5 is more complicated. However, if we have a `FindComplement` function, we can automate the process of finding a symmetry an object which generates an ideal partial SBS should have. Here, let `Quotient` be a function which returns a quotient group and a mapping from cosets to elements of the quotient group. We have the following algorithm which returns a pair of subgroups $(H, K)$ such that if $p$ has $H \leq \mathrm{Stab}_{O(3)}(p)$ and $\mathrm{Stab}_S(p) = K$ then $p$ generates an ideal partial SBS.

---

**Algorithm 2** Equivariant partial SBS from object

---

**Input**

    $S$            Symmetry of input expressed as pair $(g_S, \mathtt{name}(S))$

    $\mathrm{Cl}_S(K)$     Set of conjugate subgroups expressed as $(S, K) = ((g_S, \mathtt{name}(S)), (g_K, \mathtt{name}(K)))$

    $p$            Canonical partial symmetry breaking object

**Output**

    $P$            Symmetry breaking set expressed as a pair $(p', N)$

$N_1 \leftarrow \mathtt{Normalizers}[\mathtt{name}(S)]$

$(n, \mathtt{name}(N_2)) \leftarrow \mathtt{Normalizers}[\mathtt{name}(K)]$

$N_2 \leftarrow (g_S^{-1} g_K n, \mathtt{name}(N_2))$

$N \leftarrow N_1 \cap N_2$

$N \leftarrow (e, \mathtt{name}(S)) N$

$N \leftarrow g_S N g_S^{-1}$

**return** $(g_S p, N)$

---

**Algorithm 3** Ideal partial SBS generating object symmetry

---

**Input**

    $S$            Symmetry of input expressed as pair $(g_S, \mathtt{name}(S))$

    $\mathrm{Cl}_S(K)$     Set of conjugate subgroups expressed as $(S, K) = ((g_S, \mathtt{name}(S)), (g_K, \mathtt{name}(K)))$

**Output**

    $H$            Symmetry needed for object $p$ to generate ideal partial SBS

$N_1 \leftarrow \mathtt{Normalizers}[\mathtt{name}(S)]$

$(n, \mathtt{name}(N_2)) \leftarrow \mathtt{Normalizers}[\mathtt{name}(K)]$

$N_2 \leftarrow (g_S^{-1} g_K n, \mathtt{name}(N_2))$

$N \leftarrow N_1 \cap N_2$

$N' \leftarrow (e, \mathtt{name}(S)) \cap N_2$

$(Q_1, \phi) \leftarrow \mathtt{Quotient}[N, (g_S^{-1} g_K, K)]$

$Q_2 \leftarrow \phi(N')$

$C \leftarrow \mathtt{FindComplement}[Q_1, Q_2]$

**if** $C$ exists **then**

    $(h, \mathtt{name}(H)) \leftarrow \phi^{-1}(C)$

    **return** $((g_S h, \mathtt{name}(H)), K)$

**else**

    **return** None

**end if**

---

### 5.6 Using a symmetry breaking object

In general, one has the freedom to decide how to incorporate a symmetry breaking object into their model. This is often dependent on which equivariant architecture we apply our framework to and influences the choice of a canonical symmetry breaking object. For example, if we consider images and rotational symmetry, one natural choice is to add an additional image channel for the symmetry breaking object. In this case one would naturally choose to use an asymmetric image as the canonical symmetry breaking object. In the experiments section, we apply our framework to equivariant message passing graph neural networks (GNNs). In that case we can insert the symmetry breaking object as an additional node feature and specify the representation it transforms as.

## 6 Relation to other works

Having fully described our framework, it is useful to briefly discuss how our method relates to existing symmetry breaking works.

### 6.1 Adaption of explicit symmetry breaking methods

In Smidt et al. (2021), an additional symmetry breaking object is learned from the data. Similarly in relaxed equivariant networks, we learn additional nontrivial weights which we can often interpret as an additional symmetry breaking object (Huang et al., 2023; van der Ouderaa et al., 2022; Wang et al., 2022a;b; 2023). It is therefore natural to ask how these learned symmetry breaking objects relate to our framework.

In the case where we apply the method of Smidt et al. (2021) to a single symmetry breaking sample $x, y$, then we obtain some additional input $b$ which lets an equivariant model output $y$ when given input $x$. As long as there is an action of $G$ on the learned symmetry breaking object $b$, we can adapt it to form an equivariant SBS by taking an orbit under $N_G(S)$ (or only $N_G(S, K)$). In fact with a suitable group transformation we may simply adopt them as canonical symmetry breaking objects for Algorithms 1 and 2.

However, these learned symmetry breaking objects will in general not generate an ideal SBS. The procedure of Smidt et al. (2021) uses gradients of a $G$-invariant loss. Since the loss depends on both $x, y$, we can view the obtained symmetry breaking object as coming from some $G$-equivariant function $b = h(x, y)$. Hence, $b$ will have the symmetry of $\mathrm{Stab}_G(x) \cap \mathrm{Stab}_G(y)$ which is why it can help our equivariant model break the symmetry of input $x$. In fact, this means that we always will obtain an exact SBS in the partial symmetry breaking case. However, our Theorem 3.3 and 4.5 tells us $b$ may need additional symmetry to generate an ideal SBS which Smidt et al. (2021) does not guarantee.

### 6.2 Relation to spontaneous symmetry breaking methods

In the work of Balachandar et al. (2022), symmetry breaking is done through introducing nonequivariant components in their architecture. In particular, their symmetry detection algorithm generates a vector perpendicular to the mirror plane and hence can be modified to construct a SBS by taking positive and negative values of the vector.

The framework outlined by Kaba & Ravanbakhsh (2023) advocates for a notion of relaxed equivariance where rather than $f(gx) = gf(x)$ we instead just have $f(gx) = g'f(x)$ for some $g' \in g\mathrm{Stab}_G(x) = gS$. In Definition 2.4, we argue we should consider a set-valued function $F : X \to \mathcal{Y}$ and impose equivariance for group action on the set. By restricting to a specific member of the output set, we would obtain a function $f : X \to Y$ so that $f(x) \in F(x)$. It is not hard to show that such an $f$ satisfies the relaxed equivariance of Kaba & Ravanbakhsh (2023). This can be implemented by modifying Algorithm 1 so we pick a specific $b \in B$ rather than returning the entire SBS $B$.

Finally, a natural attempt for breaking symmetry is simply noise injection (Liu et al., 2019; Locatello et al., 2020). Noise which is isotropic with respect to group $G$ can be viewed as a SBS, however, we expect our network would need to learn to map multiple (possibly infinite) input, noise pairs to the same output. A motivation of our work is that by using knowledge of input and output symmetry, we can reduce this

degeneracy and we characterize exactly how to analyze this with Corollaries 3.6 and 4.7. In many cases, we can even have the ideal degeneracy of 1. However, we would like to point out that our theoretical results also let us prove we sometimes cannot do better than noise. For example, we can use Theorem F.1 and Corollary 3.6 to show that for $G = SO(2)$, any full symmetry breaking scheme using only input symmetry must have infinite degeneracy.

# 7 Experiments

Here, we provide some example tasks where we apply our framework of full symmetry breaking and partial symmetry breaking to an equivariant message passing GNN. We consider the cases where we can find an ideal equivariant SBS or partial SBS. This section serves primarily as a proof of concept for how our approach works in practice. Note that while these examples only have a single type of input in training for simplicity, this is by no means a restriction of our framework. In general one simply uses the given symmetry of the inputs and outputs in partial symmetry breaking in addition to the algorithms from Section 5 to choose what specific symmetry breaking object to use. This lets us work with multiple types of inputs and input symmetries with the same model.

## 7.1 Full symmetry breaking: triangular prism

### 7.1.1 Task

For an example of full symmetry breaking, we consider the task of picking a vertex of a triangular prism, similar to that described in Section 3.2.

**Input:** Graph with 6 nodes with edges given by the edges of the prism and position features at the nodes corresponding to the positions of the vertices. We add an additional pseudoscalar feature of magnitude 1.

**Input symmetry:** The prism with only the position features has $D_{3h}$ symmetry, but the addition of pseudoscalar features reduces this to $D_3$ symmetry.

**Output:** Vector which points from the center of the prism to one of the vertices. Note that there is a set of equally valid outputs.

**Output symmetry:** Nonzero vectors have $C_{\infty v}$ symmetry where the high symmetry axis is aligned along the vector.

We note that the there is no shared symmetry between the output and input if we include the pseudoscalar feature on the prism. Hence, this is an example of a full symmetry breaking task. Further, this is a spontaneous symmetry breaking task because as we rotate the prism, the set of valid vectors rotate as well.

### 7.1.2 Architecture

We apply our framework to the default equivariant message passing GNN implemented in the `e3nn` pytorch library Geiger & Smidt (2022). Each layer consists of the equivariant 3D steerable convolutions followed by gated nonlinearities described in Weiler et al. (2018). Node features in this network are separated into 3 types, position features, node features, and node attributes. The position features are used for point convolutions and node attributes are permanent features which remain through all message layers. Naturally, we input the positions of the prism vertices as position features. We input the pseudoscalar feature as a shared node attribute for all vertices. Similarly, to implement the input of a symmetry breaking object, we also add it as a shared node attribute for all vertices.

We speicify the irreps of the hidden features in our model up to $l = 2$ of both parities and use up to $l = 4$ spherical harmonics for point convolution filters. Further for the radial network, we use a 3 layer fully connected network with 16 hidden features in each layer. In the final layer, we specify a odd node $l = 1$ (vector) feature. We take the sum of the final output vectors as the model output.

### 7.1.3 Symmetry breaking set

In this case, it turns out one can set the vector $(\sqrt{3}/2, 1/2, 0)$ as a canonical symmetry breaking object and it will generate an ideal SBS. See Appendix H.1.1 for details on why this choice works. Following Algorithm 1, we see that we will obtain a set of unit vectors parallel to an edge of the triangular faces of the prism as our SBS.

### 7.1.4 Results

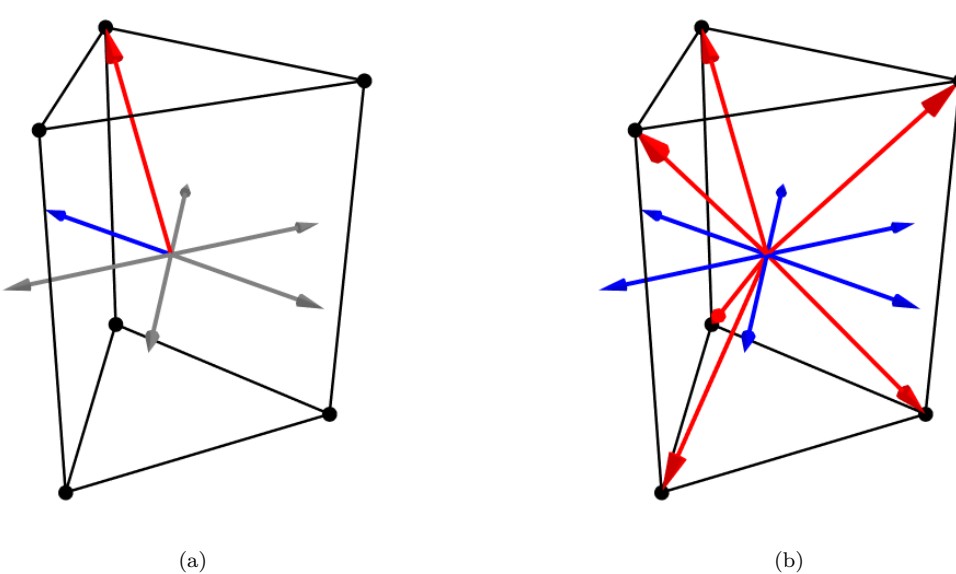

| (a) | (b) |

Figure 10: (a) Output (red) generated by our model and symmetry breaking object (blue) given. (b) The set of all the outputs generated by our model if we feed in all symmetry breaking objects.

Since our framework is equivariant, we fix the orientation of the prism in training.

We first fix a choice of one symmetry breaking object from our equivariant SBS and one of the vertices of the triangular prism. We then give the chosen symmetry breaking object as the additional input to our model and train the model to match the vector from the center of the prism to the chosen vertex using MSE loss. An example of the result of this training is shown in Figure 10a. We also observe that no matter which pair of vertex and symmetry breaking object we pick, our equivariant network is able to learn the vector pointing that that vertex. In practice, this means that we can match any of our symmetry breaking objects with a single observed output.

Once trained on one pair of symmetry breaking object and vertex, the equivariance of our GNN means that inputting the other symmetry breaking objects in our SBS gives the other symmetrically related outputs. This is shown in Figure 10b.

Further, rather than picking one vertex, we also tried modifying our loss so that we compute the MSE loss for all choices of vertex and take the minimum. Hence, our network can learn which vertex to pair with each symmetry breaking object. In this prism example, our pairing is random. This method of taking the minimum loss is especially useful when we have multiple instances of symmetry breaking in our data.

Finally, in Appendix H.1.2 we demonstrate that a non-equivariant SBS fails as described in Section 3.2 and in Appendix H.1.3 we demonstrate the degeneracy of nonideal SBS compared to an ideal one as described in Section 3.4.

## 7.2 Partial symmetry breaking: octagon to rectangle

### 7.2.1 Task

For an example of partial symmetry breaking, we consider the task of deforming an octagon to a rectangle.

**Input:** Graph with 8 nodes with edges corresponding to the edges of the octagon and position features at nodes corresponding to positions of the vertices. We add an additional pseudoscalar feature of magnitude 1.

**Input symmetry:** The octagon with only the position features has $D_{8h}$ symmetry, but the addition of pseudoscalar features reduces this to $D_8$ symmetry.

**Output:** Graph with 8 nodes with edges corresponding to the edges of the octagon and position features at nodes corresponding to new positions of the vertices. The positions are such that 4 of the vertices which form the shape of a rectangle remain unchanged and the remaining 4 vertices are shifted so their positions coincide with the nearest of the 4 vertices. Note that there is a set of equally valid outputs.

**Output symmetry:** Output graph has $D_{2h}$ symmetry.

We note that the shared symmetry between the input and output is $D_2$ compared to the $D_8$ symmetry of the input. Hence, we may apply our partial SBS framework here. The reason for forcing $D_8$ symmetry by adding chirality is to demonstrate that each step in Algorithm 3 in general is a nontrivial operation.

### 7.2.2 Architecture

We use the same default equivariant message passing GNN implemented as for the prism case. The symmetry breaking object is a different type of representation, so we specify different irreps for the shared node attribute for all vertices to input the symmetry breaking object. In addition, we keep the final output irreps as a vector at all nodes, but rather than averaging them, we now interpret them as a displacement to generate new positions for the distorted octagon.

### 7.2.3 Symmetry breaking set

In this case, choosing $(0, 0, 1, 0, 0)$ for a $l = 2$ irrep as a canonical partial symmetry breaking object generates an ideal partial symmetry breaking set. See Appendix H.2.1 The resulting SBS from Algorithm 2 is the set of $l = 2$ harmonics "parallel" to the edges of the octagon.

### 7.2.4 Results

Similar to the prism case, we try training by matching a specific symmetry breaking object to a rectangle distortion. When the symmetry breaking object and rectangle are compatible (share the same symmetries), then our model has no problem learning to deform the octagon into the rectangle. This is shown in Figures 11a and 11b. An interesting failure case occurs when we try to match a symmetry breaking object and rectangle with incompatible symmetries. This is shown in Figure 11c. Here, the $D_2$ symmetry of the rectangle and of the symmetry breaking object are misaligned. As a result, our model predicts an output which has symmetry of $D_4$ which is the group generated when we include the symmetry elements of both the target rectangle and the symmetry breaking object. Hence, the resulting shape is a square.

As with the triangular prism case, we also tried letting the model choose which rectangle to deform to given a symmetry breaking object. In this case, our model computes loss separately for all 4 possible rectangle distortions and takes the minimum. We note that for a given symmetry breaking object, 2 of the possible rectangles are symmetrically compatible while 2 are not. Over 200 random initializations, we find roughly 30% of the time our model attempts to match symmetrically incompatible symmetry breaking objects to a rectangle. This is better than the 50% we would expect if it matches pairs randomly.

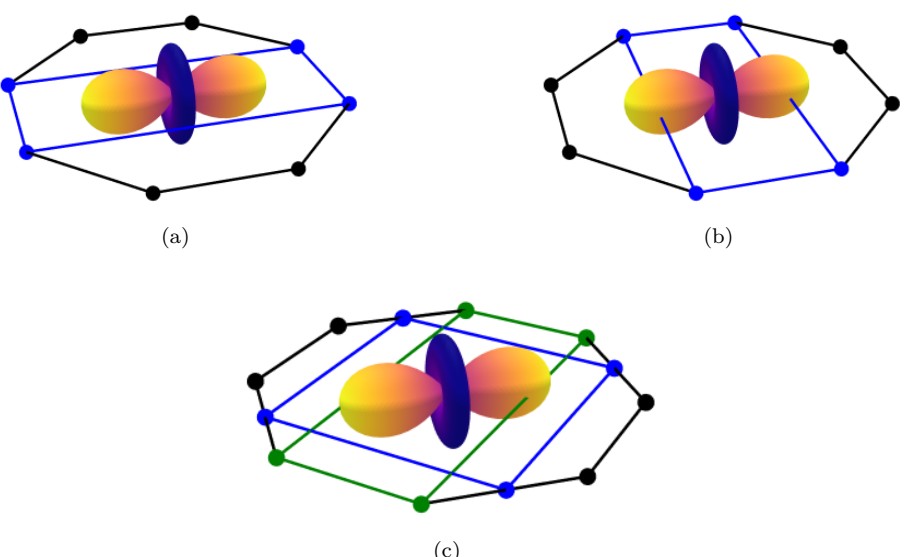

Figure 11: (a) Output (blue) of our model when we match a symmetry breaking object with a compatible rectangle. (b) Output (blue) of our model when we match a symmetry breaking object with a different compatible rectangle. (c) Output (blue) when we match a symmetry breaking object with an incompatible rectangle (green). Note the square has symmetries of both the symmetry breaking object and the target rectangle.

### 7.3 BaTiO$_3$ phase transitions

#### 7.3.1 Task

Finally, we demonstrate our framework on a more realistic example. For this, we examine the crystal structure of barium titanate (BaTiO$_3$). Specifically, as we decrease temperature, there is a phase transition from a high space-group symmetry $P_{m\bar{3}m}$ state to a lower space-group symmetry $P_{4mm}$ state at 403K Kay & Vousden (1949); Oliveira et al. (2020); Woodward (1997). The high and low symmetry states are shown in Figures 12a and 12b respectively. Note that the real distortions are rather small and hard to see visually. Table 1 provides some numerical quantities which help distinguish the two. In particular, there are 3 distinct Ti-O-Ti bond angles in a primitive cell, 2 of which are distorted equally to 171.80° in the low symmetry structure. This bent angle is shown more clearly in the schematic in Figure 12b.

**Input:** Graph with 5 nodes corresponding to the atoms in a primitive cell of BaTiO$_3$. We have position features corresponding to the positions of the atoms in the cell. We fix the unit cell to be a cube with sides 4Å, close to the size of the real cells.

**Input symmetry:** The high symmetry structure has a space group symmetry of $Pm\bar{3}m$. Ignoring translational symmetries, this corresponds to $O_h$ point group symmetry (denoted as $m\bar{3}m$ in Hermann-Maugin notation). Note the symmetry is already listed in the materials project database Jain et al. (2013).

**Output:** Graph with 5 nodes corresponding to the atoms in a primitive of BaTiO$_3$. We have position features corresponding to the positions of the atoms in the cell. We fix the unit cell to be a cube with sides 4Å, close to the size of the real cells.

**Output symmetry:** The high symmetry structure has a space group symmetry of $P4mm$. Ignoring translational symmetries, this corrsponds to $C_{4v}$ point group symmetry (denoted as $4mm$ in Hermann-Maugin notation). Note the symmetry is already listed in the materials project database Jain et al. (2013).

The input and output share $D_{4h}$ symmetry compared to the $O_h$ symmetry of the input. Hence may apply our partial symmetry breaking framework here.

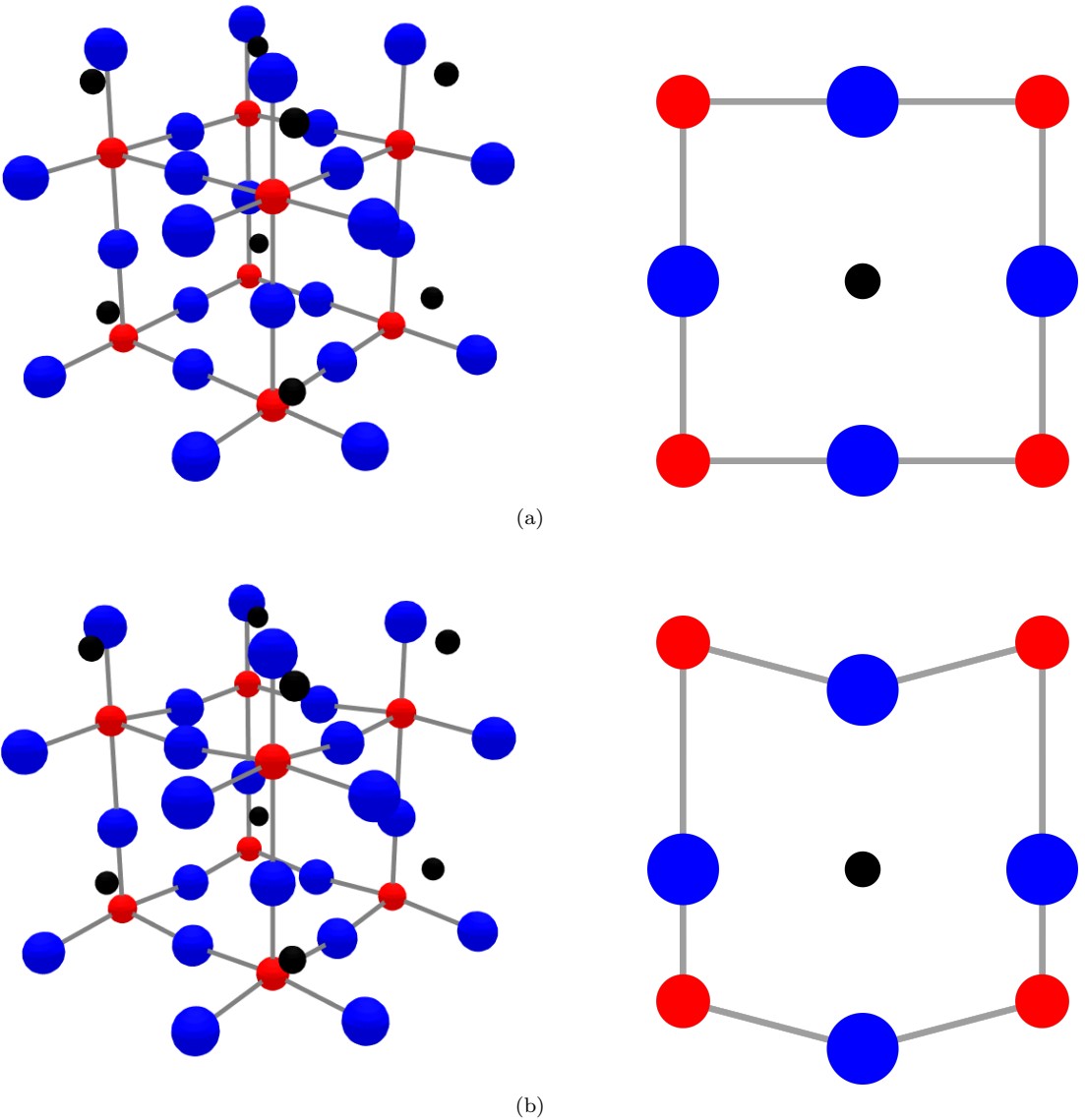

Figure 12: (a) Initial high symmetry crystal structure of BaTiO$_3$. Left is an actual plot of the crystal structure and right is a side-on schematic. (b) Target low symmetry crystal structure of BaTiO$_3$. Left is an actual plot of the distorted crystal structure and right is a side-on schematic with exaggerated distortion. The angle of the bent bond is $171.80°$.

### 7.3.2 Architecture

We use the same default equivariant message passing GNN implemented as for the previous examples. Here, we also input atom type as an additional node feature using one hot encoding. Similar to the octagon example, have vector features at each node as output and interpret it as the displacement to generate the distorted structure. We can choose a SBS consisting of a single vector so we have an additional vector as node attribute to input a symmetry breaking object. Finally, since crystals are periodic, we modified the network to include periodicity when computing relative displacement vectors.

### 7.3.3    Symmetry breaking set

Note that $O_h$ has itself as normalizer in $O(3)$ so the symmetry completely determines orientation. Hence any object sharing $C_{4v}$ symmetry works for generating an ideal equivariant partial SBS. A simple choice consists of vectors (odd parity $l = 1$ object) pointing along the 4-fold rotation axes.

### 7.3.4    Results

As shown in Figure 13 and Table 1, our model is able to learn to distort the crystal structure appropriately when given an appropriate symmetry breaking object. Without such an input the model cannot provide any distortions. In additional, we computed some additional invariant quantities of the crystal structures and we see that these invariants for the output structure of the model with a SB object matches the invariants for the target lower symmetry structure.

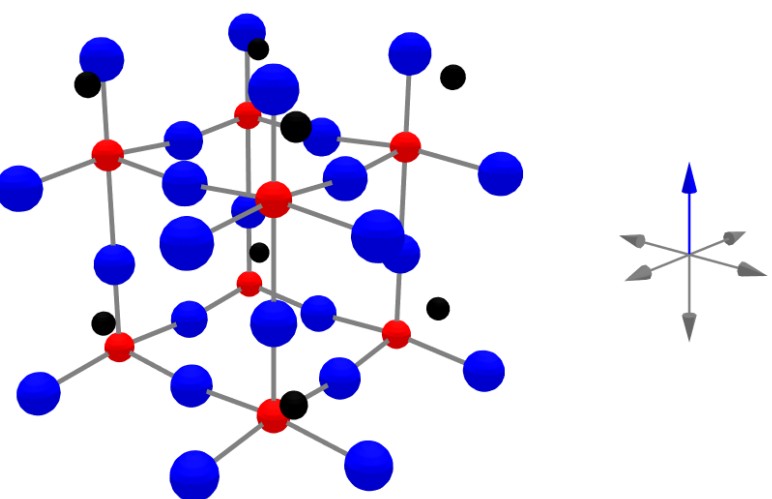

Figure 13: Distorted crystal structure generated by our model when given a symmetry breaking object shown on the right in blue.

Table 1: Values of various quantities which help distinguish the high symmetry and low symmetry structures. Our models here try to distort the high symmetry structure to the low symmetry one.

| Structure | Bond length average | Bond length variance | Ti-O-Ti |
|---|---|---|---|
| High symmetry | 2 | 0 | 180° |
| Low symmetry | 2.003417 | 0.01392 | 171.80° |
| Model (no SBS) | 2 | 0 | 180° |
| Model (SB object $(1, 0, 0)$) | 2.003417 | 0.01392 | 171.80° |

## 8    Conclusion

We formalize the problem equivariant neural networks face in the spontaneous symmetry breaking setting. We propose the idea of equivariant symmetry breaking sets which allows ENNs to sample or generate all possible symmetrically related outputs given a highly symmetric input. Importantly, we show that minimizing these sets is intimately connected to a well studied group theory problem, and tabulate solutions for the ideal case for the point groups. We then demonstrate how our symmetry breaking framework works in practice on example problems.

One future direction is to include translations and tabulate complements for the space groups in their respective normalizers. This would be particularly useful for crystallography applications. Another direction is to automate finding stabilizers for partial symmetry breaking objects. In addition, our method assumes we can efficiently detect the symmetry of our input and outputs. Designing fast symmetry detection algorithms would also be extremely beneficial. Finally, designing efficient loss functions which do not punish symmetrically related outputs would be useful for any network dealing with spontaneous symmetry breaking.

### Acknowledgments

We thank the helpful discussions with Robin Walters, Elyssa Hofgard, and Rui Wang for framing the different types of symmetry breaking.

YuQing Xie was supported by the MIT College of Computing fellowship and the National Science Foundation Graduate Research Fellowship under Grant No. DGE-1745302. Tess Smidt was supported by DOE ICDI grant DE-SC0022215.

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

## Table of Contents

# A    Notation and commonly used symbols

Here, we present the notation we use throughout this paper and the typical variable names.

Table 2: Notation used throughout this paper

| | |
|---|---|
| $\text{Stab}_G(x)$ | Stabilizer of an element $x$ under a group $G$ |
| $N_G(S)$ | Normalizer of group $S$ in group $G$ |
| $\text{Cl}_G(S)$ | Set of groups obtained by conjugating group $S$ with elements in $G$ |
| $\text{Orb}_G(x)$ | Orbit of an element $x$ under action by elements of group $G$ |
| $\mathcal{P}(X)$ | Set of all subsets of $X$ |
| $G/S$ | When $G$ is a group, this is the set of left cosets. If $S$ is a normal subgroup, this also denotes the quotient group |
| $X/S$ | When $X$ is a set, this is the equivalence classes induced by action of $S$ on $X$ |
| $S \leq G$ | If $S$ and $G$ are groups, this denotes that $S$ is a subgroup of $G$ |
| $f|_X$ | Function $f$ with domain restricted to $X$ |

Table 3: Commonly used symbols

| | |
|---|---|
| $G$ | Group our network is equivariant under |
| $\mathbf{1}$ | Used to denote the trivial group |
| $e$ | Identity element of a group |
| $x$ | Input |
| $y$ | Output |
| $S$ | Symmetry of our input, more precisely $\text{Stab}_G(x)$ |
| $K$ | Symmetry of our output, more precisely $\text{Stab}_S(y)$ |
| $B$ | Full symmetry breaking set |
| $P$ | Partial symmetry breaking set |

# B    Group theory

Group theory is the mathematical language used to describe symmetries. Here, we present a brief overview of concepts from group theory we need to both define equivariance, and to understand our proposed symmetry breaking scheme. For a more comprehensive treatment of group theory, we refer to standard textbooks Dresselhaus et al. (2007); Dummit & Foote (2004); Kurzweil & Stellmacher (2004). We begin by defining what a group is.

**Definition B.1** (Group)**.** Let $G$ be a nonempty set equipped with a binary operator $\cdot : G \times G \to G$. This is a group if the follwing group axioms are satsfied

1. Associativity: For all $a, b, c \in G$, we have $(a \cdot b) \cdot c = a \cdot (b \cdot c)$

2. Identity element: There is an element $e \in G$ such that for all $g \in G$ we have $e \cdot g = g \cdot e = g$

3. Inverse element: For all $g \in G$, there is an inverse $g^{-1} \in G$ such that $g \cdot g^{-1} = g^{-1} \cdot g = e$ for identity $e$.

Some examples of groups include the group of rotation matrices with matrix multiplication as the group operation, the group of integers under addition, and the group of positive reals under multiplication. One very important group is the group of automorphisms on a vector space. This group is denoted $GL(V)$ and we can think of it as the group of invertible matrices.

While abstractly groups are interesting on their own, we care about using them to describe symmetries. Intuitively, the group elements abstractly represent the symmetry operations. In order to understand what these actions are, we need to define a group action.

**Definition B.2** (Group action)**.** Let $G$ be a group and $\Omega$ a set. A group action is a function $\alpha : G \times \Omega \to \Omega$ such that $\alpha(e, x) = x$ and $\alpha(g, \alpha(h, x)) = \alpha(gh, x)$ for all $g, h \in G$ and $x \in \Omega$.

Often, we may want to relate two groups to each other. This is done using group homomorphisms, a mapping which preserves the group structure.

**Definition B.3** (Group homomorphism and isomorphism)**.** Let $G$ and $H$ be groups. A group homomorphism is a function $f : G \to H$ such that $f(u \cdot v) = f(u) \cdot f(v)$ for all $u, v \in G$. A group homomorphism is an isomorphism if $f$ is a bijection.

Because there are many linear algebra tools for working with matrices, it is particular useful to relate arbitrary groups to groups consisting of matrices. Such a homomorphism together with the vector space the matrices act on is a group representation.

**Definition B.4** (Group representation)**.** Let $G$ be a group and $V$ a vector space over a field $F$. A group representation is a homomorphism $\rho : G \to GL(V)$ taking elements of $G$ to autmorphisms of $V$.

Given any representation, there are often orthogonal subspaces which do not interact with each other. If this is the case, we can break our representation down into smaller pieces by restricting to these subspaces. Hence, it is useful to consider the representations which cannot be broken down. These are known as the irreducible representations (irreps) and often form the building blocks of more complex representations.

**Definition B.5** (Irreducible representation)**.** Let $G$ be a group, $V$ a vector space, and $\rho : G \to GL(V)$ a representation. A representation is irreducible if there is no nontrivial proper subspace $W \subset V$ such that $\rho|_W$ is a representation of $G$ over space $W$.

There has been much work on understanding the irreps of various groups and many equivariant neural network designs use this knowledge.

One natural question is whether there is a subset of group elements which themselves form a group under the same group operation. Such a subset is a called a subgroup.

**Definition B.6** (Subgroup)**.** Let $G$ be a group and $S \subseteq G$. If $S$ together with the group operation of $G$ $\cdot$ satisfy the group axioms, then $S$ is a subgroup of $G$ which we denote as $S \leq G$.

One particular feature of a subgroup is that we can use them to decompose our group into disjoint chunks called cosets.

**Definition B.7** (Cosets)**.** Let $G$ be a group and $S$ a subgroup. The left cosets are sets obtained by multiplying $S$ with some fixed element of $G$ on the left. That is, the left cosets are for all $g \in G$

$$gS = \{gs : s \in S\}.$$

We denote the set of left cosets as $G/S$. The right cosets are defined similarly except we multiply with a fixed element of $G$ on the right. That is, the right cosets are for all $g \in G$

$$Sg = \{sg : s \in S\}.$$

We denote the set of right cosets as $G\backslash S$.

In general, the left and right cosets are not the same. However, for some subgroups they are the same. Those subgroups are called normal subgroups.

**Definition B.8** (Normal subgroup)**.** Let $G$ be a group and $N$ a subgroup. Then $N$ is a normal subgroup if for all $g \in G$, we have $gNg^{-1} = N$.

It turns out that given a normal subgroup, one can construct a group operation on the cosets. The resulting group is called a quotient group.

**Definition B.9** (Quotient group)**.** Let $G$ be a group and $N$ a normal subgroup. One can define a group operation on the cosets as $aN \cdot bN = (a \cdot b)N$. The resulting group is called the quotient group and is denoted $G/N$.

For subgroups $S$ which are not normal in $G$, it is often useful to consider a subgroup of $G$ containing $S$ where $S$ is in fact normal. The largest such subgroup is called the normalizer.

**Definition B.10** (Normalizer)**.** Let $G$ be a group and $S$ a subgroup. The normalizer of $S$ in $G$ is

$$N_G(S) = \{g : gSg^{-1} = S\}.$$

Similar to orthogonal vector spaces, one can imagine an analogous notion for groups. These are called complement subgroups.

**Definition B.11** (Complement)**.** Let $G$ be a group and $S$ a subgroup. A subgroup $H$ is a complement of $S$ if for all $g \in G$, we have $g = sh$ for some $s \in S$ and $h \in H$ and $S \cap H = \{e\}$.

It turns out that if $S$ is a normal subgroup of $G$ and $H$ is a complement, then $H$ is isomorphic to the quotient group.

Finally, it is useful to define what we mean by symmetry of an object. These are all group elements which leave the object unchanged and is called the stabilizer.

**Definition B.12** (Stabilizer)**.** Let $G$ be a group, $\Omega$ some set with an action of $G$ defined on it, and $u \in \Omega$. The stabilizer of $u$ is all elements of $G$ which leave $u$ invariant. That is

$$\mathrm{Stab}_G(u) = \{g : gu = u, g \in G\}.$$

One can check that the stabilizer is indeed a subgroup. Closely related to the stabilizer is the orbit. This is all the values we get when we act with our group on some object.

**Definition B.13** (Orbit)**.** Let $G$ be a group, $\Omega$ some set with an action of $G$ defined on it, and $u \in \Omega$. The orbit of $u$ is the set of all values obtained when we act with all elements of $G$ on it. That is,

$$\mathrm{Orb}_G(u) = \{gu : g \in G\} = Gu.$$

It turns out one can show that the stabilizer of elements in the orbit are related. This relation turns out to be conjugation which we define below.

**Definition B.14** (Conjugate subgroups)**.** Let $S$ and $S'$ be subgroups of $G$. We say $S$ and $S'$ are conjugate in $G$ if there is some $g \in G$ such that $S = gS'g^{-1}$. We denote the set of all conjugate subgroups by

$$\mathrm{Cl}_G(S) = \{gSg^{-1} : g \in G\}.$$

# C   Equivariant neural networks

Here, we give a brief overview of equivariant neural networks. For a more in depth coverage of the general theory and construction of equivariance, we refer to works such as Cohen et al. (2019); Finzi et al. (2020); Kondor & Trivedi (2018). We emphasize that the symmetry breaking techniques presented in the paper apply to any equivariant architecture.

We first define equivariance.

**Definition C.1** (Equivariance)**.** Let $G$ be a group with actions on spaces $X$ and $Y$. A function $f : X \to Y$ is said to be equivariant if for all $x \in X$ and $g \in G$ we have

$$f(gx) = gf(x).$$

Intuitively, we can interpret this as rotating the input giving the same result as just rotating the output. It is easy to check that the composition of equivariant functions is still an equivariant function. Hence, equivariant neural networks are designed using a composition of equivariant layers.

There has been considerable study into how one should design equivariant layers. One approach is to modify convolutional filters by transforming them with the elements of our group Cohen & Welling (2016a). This approach is known as group convolution and is based on the intuition that convolutional filters are translation equivariant. In group convolution, one interprets our data as a signal over some domain. The first layer is a lifting convolution which transforms our data into a signal over the group. The remaining layers then just convolve this signal with filters which are also signals over the group.

One can further use group theory tools to break down the convolutional filters into irreps. This leads to steerable convolutional networks Cohen & Welling (2016b). These can be extended and used to parameterize continuous filters which can be used for infinite groups Cohen et al. (2018). It turns out the irreps of the group are natural data types for equivariant networks. Further, we can express the convolutions as tensor products of irreps. We can think of equivariant operations as being composed of tensor products of irreps, linear mixing of irreps, and scaling by invariant quantities. Combining these, we get tensorfield networks which works on point clouds and is rotation equivariant Thomas et al. (2018). In this paper, we demonstrate our method using networks built from the `e3nn` framework for $O(3)$ equivariance Geiger & Smidt (2022).

## D Limitations

### D.1 Symmetry detection

To use our procedure, we do assume knowledge of the symmetries of the inputs and outputs to our network. In the full symmetry breaking framework, we only need the symmetry of the input. In the partial symmetry breaking framework, we need the symmetry of both the input and the output. However, we argue that this is not a major concern.

First, symmetry detection is a well studied problem and there are many algorithms exist for various types of data Bokeloh et al. (2009); Keller & Shkolnisky (2004); Largent et al. (2012); Mitra et al. (2006). Further, sometimes the symmetries of the inputs and outputs are already known. This is especially true for crystallographic data Jain et al. (2013). In addition, because we only need the symmetry to design equivariant SBSs, we only need to perform symmetry detection once. This can simply be incorporated as a preprocessing step for our data.

Further, we want to emphasize that our framework can be used to prove whether knowledge of input symmetry is beneficial. We prove in Lemma F.1 that no finite subgroups of $SO(2)$ have complement in their normalizer (which is also just $SO(2)$). Combined with Corollary 3.6 this actually implies the degeneracy of any $SO(2)$-equivariant SBS for cyclic groups is infinite. Hence, we cannot do much better than a something like noise injection, which introduces asymmetry without knowledge of input symmetry.

### D.2 Loss functions

While this work focuses on allowing equivariant networks to produce a set of lower symmetry outcomes, it turns out another important problem is designing appropriate loss functions. Suppose we only have one example input output pair $x, y$ where $y$ shares no symmetries with $x$. In this case there is no problem. We can fix any $b \in B$ and train to minimize a simple MAE loss $||f(x, b) - y||^2$ for example.

However, suppose we observe $x, y$ and $x, y'$ in the data where $y' = sy$ and $s \in S = \text{Stab}_G(x)$. Then suppose we try to minimize MSE loss $||f(x, b) - y||^2 + ||f(x, b') - y'||^2$, where $b' = s'b$. Then by equivariance, the second term in the loss is

$$||f(x, b') - y'||^2 = ||f(x, s'b) - y'||^2 = ||s'f(x, b) - sy||^2 = ||f(x, b) - s'^{-1}sy||^2.$$

So in fact we see we must choose $s' = s$. So with multiple input output pairs and a simple loss directly comparing outputs such as MSE, we have a problem pairing symmetry breaking objects with outputs.

However, suppose instead our loss was chosen such that $\texttt{loss}(y, y') = \texttt{loss}(y, y)$ is small if $y' = sy$ for any $s \in S$. Then even if our network outputs $sy$ instead of $y$ when given some symmetry breaking object $b$, we do not punish it. Hence, this pairing problem would not exist. A simple version of such a loss would be to compute the MSE for all possible symmetrically related outputs and take the closest one

$$\texttt{loss}(f(x, b), y) = \min_{s \in S}(||f(x, b) - sy||^2).$$

This is what we use for our experimental examples in this work. However, this can be inefficient for large or infinite $S$ and designing appropriate loss functions in such cases remains an open question.

# E    Proofs

## E.1    Proof of Lemma 2.1

**Lemma E.1.** *Let $X$ and $Y$ be spaces equipped with a group action of $G$. Suppose the action on $X$ is transitive. We can choose a $G$-equivariant $f : X \to Y$ such that $f(u) = y$ if and only if $\mathrm{Stab}_G(y) \geq \mathrm{Stab}_G(u)$. Further this uniquely defines $f$.*

*Proof.* First suppose we did have $f(u) = y$. For any $g \in \mathrm{Stab}_G(u)$, we have by equivariance of $f$ that

$$gy = f(gu) = f(u) = y.$$

So $g \in \mathrm{Stab}_G(y)$.

Next, suppose $\mathrm{Stab}_G(y) \geq \mathrm{Stab}_G(u)$. For any $x \in X$, there is some $r \in G$ so that $x = ru$. Let us pick exactly one such $r$ for each $X$ and form a set $R$. Hence any $x$ is uniquely written as $x = ru$ for $r \in R$. Define

$$f(x) = f(ru) = ry.$$

We claim $f$ is equivariant. For any $g \in G$ and $x \in X$, let $x = ru$ and $gx = r'u$ for some $r, r' \in R$. Then,

$$f(gx) = f(gru) = f(r'u) = r'y.$$

But note that $gx = r'u$ implies $r'^{-1}gx = r'^{-1}gru = u$. So $r'^{-1}gr \in \mathrm{Stab}_G(u) \leq \mathrm{Stab}_G(y)$. Hence, we also have $r'^{-1}gry = y$. So,

$$f(gx) = r'y = r'(r'^{-1}gry) = gry = gf(x).$$

Hence, $f$ is equivariant.

Finally, for uniqueness, suppose $f, f'$ are two equivariant functions such that $f(u) = f'(u) = y$. Then by equivariance, for any $x = gu \in X$ we have

$$f(x) = f(gu) = gy = f'(gu) = f'(x).$$

$\square$

## E.2    Formal justification of Definition 3.2

We can justify Definition 3.2 by characterizing exactly when $\sigma$ can be equivariant. This leads to the following proposition.

**Proposition E.2.** *Let $G$ be a group and $S$ be a subgroup of $G$. Let $B \in \mathcal{P}(\mathbf{B})$ be a set where there is some group action of $G$ defined on $\mathbf{B}$. Then there exists an equivariant $\sigma|_{\mathrm{Cl}_G(S)} : \mathrm{Cl}_G(S) \to \mathcal{P}(\mathbf{B})$ such that $\sigma|_{\mathrm{Cl}_G(S)} = B$ if and only if $nB = B$ for all $n \in N_G(S)$.*

*Proof.* Note that $\mathrm{Cl}_G(S)$ is a set where action by conjugation is a transitive one. Also note by definition that $\mathrm{Stab}_G(S)$ for this action is precisely the definition of a normalizer $N_G(S)$. Then by Lemma E.1, we see such a function exists if and only $B$ is also symmetric under $N_G(S)$. $\square$

## E.3    Proof of Theorem 3.3

**Theorem 3.3.** *Let $G$ be a group and $S$ a subgroup. Let $B$ be a $G$-equivariant SBS for $S$. Then it is possible to choose an ideal $B$ if and only if $S$ has a complement in $N_G(S)$.*

*Proof.* Suppose $B$ is transitive under $S$ and pick $b \in B$. Consider the stabilizer group $\mathrm{Stab}_{N_G(S)}(b)$. For any $g \in N_G(S)$, by transitivity under $S$ we must have $gb = sb$ for some $s \in S$. So, $s^{-1}gu = u$ implying that $h = s^{-1}g \in \mathrm{Stab}_{N_G(S)}(b)$. So we find that we can write any $g$ as $g = sh$ for some $s \in S$ and $h \in \mathrm{Stab}_{N_G(S)}(u)$ so

$$N_G(S) = S \cdot \mathrm{Stab}_{N_G(S)}(u).$$

But note that since $B$ is symmetry breaking, $S \cap \mathrm{Stab}_{N_G(S)}(u) = \{e\}$. Hence, $\mathrm{Stab}_{N_G(S)}(u)$ is indeed a complement.

For the converse, suppose $H$ is a complement of $S$ in $N_G(S)$. We claim $B = N_G(S)/H$ is the equivariant SBS we desire. Note that clearly by construction, this is closed under $N_G(S)$ so we satisfy the equivariance condition. Further, note that $\mathrm{Stab}_S(H) = \mathrm{Stab}_{N_G(S)}(H) \cap S = H \cap S = \{e\}$. Since $B$ is transitive under $N_G(S)$, stabilizers of all other elements are obtained by conjugation and hence also trivial. Hence, it is indeed symmetry breaking. Finally, any $g \in N_G(S)$ is uniquely written as $sh$ for some $s \in S, h \in H$ so $gH = shH = sH$. So $B$ is transitive under $S$ as well. $\qquad\square$

### E.4 Proof of Corollary 3.6

**Corollary 3.6.** *Let $G$ be a group and $S$ a subgroup. Let $M$ be such that $S \leq M \leq N_G(S)$. Let $B$ be a $G$-equivariant SBS for $S$ which is transitive under $N_G(S)$. Then it is possible to choose $B$ such that every $M$-orbit is also transitive under $S$ if and only if $S$ has a complement in $M$. In particular, such a $B$ has*

$$\mathrm{Deg}_S(B) \leq |N_G(S)/M|.$$

*Proof.* Suppose we have such a $B$ and pick any $b \in B$. By transitivity of the orbit under $S$, we have $Mb = Sb$. Let $B' = Mb$. We can check that this is in fact an ideal $M$-equivariant SBS for $S$. That it is a symmetry breaker follows since $B$ is symmetry breaking. That it is $M$-equivariant and transitive follows since $Mb = Sb$ and $N_M(S) = M$. By Theorem 3.3 this implies $S$ has a complement in $M$.

Next, suppose we have a complement of $S$ in $M$. By Theorem 3.3 we can construct $B'$ which is an ideal $M$-equivariant SBS for $S$. We can lift this to a $G$-equivariant SBS for $S$ by just taking $B = N_G(S)B'$.

Finally, to compute the order, we note that every $S$-orbit is also a $M$ orbit. Since $B$ is transitive under $N_G(S)$, there are at most $|N_G(S)/M|$ number of $M$-orbits and hence only that many $S$-orbits. So

$$\mathrm{Deg}_S(B) \leq |N_G(S)/M|.$$

$\qquad\square$

### E.5 Justification of Definition 4.4

Similar to the full SBS case, we can justify Definition 4.4 by characterizing exactly when an equivariant $\pi$ can exist. This leads to the following proposition.

**Proposition E.3.** *Let $G$ be a group, $S$ a subgroup of $G$, and $K$ a subgroup of $S$. Let $\mathbf{P}$ be a set with a group action of $G$ defined on it and $P \subset \mathbf{P}$. There exists an equivariant $\pi|_{\mathrm{Orb}_G((S,\mathrm{Cl}_S(K)))} : \mathrm{Orb}_G((S, \mathrm{Cl}_S(K))) \to \mathcal{P}(\mathbf{P})$ such that $\pi|_{\mathrm{Orb}_G((S,\mathrm{Cl}_S(K)))}((S, \mathrm{Cl}_S(K))) = P$ if and only if $N_G(S,K)$ leaves $P$ invariant.*

*Proof.* By Lemma E.1, we need $P$ to be closed under the stabilizer of the input. But the generalized normalizer $N_G(S,K)$ is precisely this stabilizer. $\qquad\square$

### E.6 Proof of Theorem 4.5

**Theorem 4.5.** *Let $G$ be a group and $S$ and $K$ be subgroups $K \leq S \leq G$. Let $P$ be a $G$-equivariant $K$-partial SBS. Then we can choose an ideal $P$ (exact and transitive under $S$) if and only if $N_S(K)/K$ has a complement in $N_{N_G(S,K)}(K)/K$.*

*Proof.* Let $P = Su$ where $u$ has symmetry $\mathrm{Stab}_S(u) = K$. We can define an action of any coset $N_G(S,K)/K$ on $u$ as just the action of a coset representative on $u$. This is consistent since $u$ is invariant under $K$. In particular, note that $K$ is a normal subgroup of $N_S(K)$ so $N_S(K)/K$ is a quotient group. Let $B' = (N_S(K)/K)u$. Since $u$ is in a $K$-partial SBS, we must have $su \neq u$ for any $s \in S - K$. Hence, for any coset $gK \in N_S(K)/K$, $gu \neq u$ if $g \notin K$. Therefore, $B'$ must be a SBS for $N_S(K)/K$.

Next, consider any coset $gK$ in $N_{N_G(S,K)}(K)/K$. Then we know $gu \in Su$ so $gu = su$ for some $s \in S$. Since $K$ was a symmetry of $u$, $gKg^{-1} = sKs^{-1}$ is a symmetry of $gu = su$. So the stabilizer of $su$ must be $sKs^{-1} = K$. Hence, $s$ must be in $N_S(K)$. Therefore the action of $gK$ on $u$ gives us an element of $B' = (N_S(K)/K)u$. Hence $B'$ is $N_{N_G(S,K)}(K)/K$-equivariant.

By Theorem 3.3, the existence of an ideal $N_{N_G(S,K)}(K)/K$-equivariant SBS for $N_S(K)/K$ implies that $N_S(K)/K$ has a complement in $N_{N_G(S,K)}(K)/K$.

For the converse direction, suppose that $A$ is a complement of $N_S(K)/K$ in $N_{N_G(S,K)}(K)/K$. Note the elements of $A$ are cosets of $K$ so we can define a set of elements of $N_G(S,K)$ as

$$H = \bigcup_{C \in A} C.$$

Define $P = \mathrm{Orb}_S(H)$. We claim that $P$ is a transitive exact equivariant partial SBS.

We first show that $P$ is exact $K$-partial symmetry breaking. Consider $s \in S$. We can write

$$sH = \bigcup_{C \in A} sC = \bigcup_{C \in sA} C.$$

Now we see if $s \in K$, then since $K$ is the identity in the quotient group $sA = A$. Hence $sH = H$ in this case. If $s \in N_S(K) - K$, then $sK$ is not the identity in $N_S(K)/K$. But $A$ is a complement so $sA \neq A$ implying $sH \neq H$. Finally, if $s \notin N_S(K)$ then $sK \notin N_{N_G(S,K)}(K)/K$. So $sK \not\subset N_{N_G(S,K)}(K)$. But $H \subset N_{N_G(S,K)}(K)$ so $sH \neq H$. Hence, $\mathrm{Stab}_S(H) = K$ and since the rest of $P$ is just the orbit of $H$, stabilizers of the other elements are in $\mathrm{Cl}_S(K)$. Hence, $P$ as we constructed is an exact $K$-partial SBS.

For equivariance consider any $n \in N_G(S,K)$ giving a coset

$$nH = \bigcup_{C \in A} nC = \bigcup_{C \in nA} C.$$

If $n \in N_{N_G(S,K)}(K)$ then since $A$ is a complement, $(nK) = (sK)(aK)$ for some $sK \in N_S(K)/K$ and $aK \in A$, so $nA = (nK)A = (sK)(aK)A = (sK)A = sA$ for some $s \in N_S(K) \subset S$. Hence, $nH = sH$ for some $s \in S$. If $n \notin N_{N_G(S,K)}(K)$, then there is some $s$ so that $nKn^{-1} = sKs^{-1}$. Therefore, $s^{-1}nKn^{-1}s = K$ so $s^{-1}n \in N_{N_G(S,K)}$. But we saw before that this means there is some $s'$ such that $s^{-1}nH = s'H$. Thus, $nH = ss'H$ and $ss' \in S$. So $nH \in P$ so $P$ is indeed closed under action by $N_G(S,K)$. $\square$

### E.7 Proof of Corollary 4.7

**Corollary 4.7.** *Let $G$ be a group, $S$ a subgroup, and $K$ a subgroup of $S$. Let $K'$ be a subgroup of $K$ and $M$ a subgroup of $N_G(S,K) \cap N_G(S,K')$ which contains $S$. Suppose $P$ is a $G$-equivariant $K$-partial SBS for $S$ which is transitive under $N_G(S,K)$. We can choose $P$ such that $\mathrm{Stab}_S(p) \in \mathrm{Cl}_{N_G(S,K)}(K')$ for all $p$ and all $M$-orbits in $P$ are transitive under $S$ if and only if $N_S(K')/K'$ has a complement in $N_M(K')/K'$. Further, such a $P$ has*

$$\mathrm{Deg}_{S,K}(P) \leq |K/K'| \cdot |N_G(S,K)/M|.$$

*Proof.* Suppose we had such a $P$. Pick some $p \in P$ such that $\mathrm{Stab}_S(p) = K'$. Since $M$-orbits are transitive under $S$, we have $Mp = Sp$. Let $P' = Mp$. We can check then that this is a $M$-equivariant set. Further, since $M \subset N_G(S,K')$, we see that this is an exact $K'$-partial symmetry breaking set. Hence, it is an ideal $M$-equivariant $K'$-partial SBS. Also note that since $M \subset N_G(S,K')$, we have $M = N_M(S,K')$. So by Theorem 4.5, $N_S(K')/K'$ must have a complement in $N_M(K')/K'$.

Conversely, suppose $N_S(K')/K'$ has a complement in $N_M(K')/K'$. Again, we note $M = N_M(S,K')$ so by Theorem 4.5 we have an ideal $M$-equivariant $K'$-partial SBS for $S$. We can lift this to $G$-equivariance by taking the orbit under $N_G(S,K)$.

To see the order of such a $P$, we consider the $S$-orbits. Let $T$ be a transversal of $S/K$. For each $S$-orbit, we can pick some $p$ in that orbit so that $\mathrm{Stab}_S(p) \leq K$. We put the elements of $Kp$ in our $P_t$. Within

each $S$-orbit, any $p'$ can be written as $sp$ for some $s \in S$ and any $s$ is uniquely written as $s = tk$ for some $t \in T$ and $k \in K$. So $p' = tkp$. However, note that any other $s'$ where $p' = s'p = sp$ can be written as $s' = sk'$ for some $k' \in \mathrm{Stab}_S(p)$. Hence, $p'$ is uniquely written as $p' = t(kp)$ since $kk'p = kp$. So each $S$-orbit contributes $|K/\mathrm{Stab}_S(p)|$ elements to $P_t$. However, since $\mathrm{Stab}_S(p) \in \mathrm{Cl}_{N_G(S,K)}(K')$, we must have $|K/\mathrm{Stab}_S(p)| = |K/K'|$. Finally, we know each $S$-orbit is also an $M$-orbit, since $P$ is transitive under $N_G(S, K)$, there are at most $|N_G(S, K)/M|$ different $S$-orbits. So

$$\mathrm{Deg}_{S,K}(P) = |P_t| \le |K/K'| \cdot |N_G(S, K)/M|.$$

$\square$

### E.8 Proof of Lemma 5.1

**Lemma 5.1.** *We have the following formula*

$$N_G(S, K) = S(N_G(S) \cap N_G(K)).$$

*Proof.* For any $n \in N_G(S, K)$, by definition we must have $nKn^{-1} = sKs^{-1}$ for some $s \in S$. Hence, $s^{-1}nKn^{-1}s = K$ so $s^{-1}n \in N_G(K)$. Further, clearly $s \in N_G(S)$ and $n \in N_G(S)$ so also $s^{-1}n \in N_G(S)$. Therefore, $s^{-1}n \in N_G(S) \cap N_G(K)$. So $n = s(s^{-1}n) \in S(N_G(S) \cap N_G(K))$. Hence

$$N_G(S, K) \subseteq S(N_G(S) \cap N_G(K)).$$

Next, consider any $n' = sn \in S(N_G(S) \cap N_G(K))$ where $s \in S, n \in N_G(S) \cap N_G(K)$. Since $s \in N_G(S), n \in N_G(S)$ clearly $sn \in N_G(S)$. Further, we find

$$snK(sn)^{-1} = s(nKn^{-1})s^{-1} = sKs^{-1} \in \mathrm{Cl}_S(K).$$

Therefore by definition $sn \in N_G(S, K)$. So also

$$S(N_G(S) \cap N_G(K)) \subseteq N_G(S, K).$$

Hence, we find that

$$N_G(S, K) = S(N_G(S) \cap N_G(K)).$$

$\square$

## F  Classification of full symmetry breaking cases for $O(3)$

Here we tabulate the cases for full symmetry breaking for the finite subgroups of $O(3)$. These are the point groups and the normalizers are tabulated in the International Tables for Crystallography in Hermann–Mauguin notation Koch & Fischer (2006). We have translated these to Schönflies notation in Table 4.

Table 4: Normalizers of the point groups in Schönflies notation. Note we have the equivalences $C_1 = 1$, $S_2 = C_i$, $C_{1h} = C_{1v} = C_s$, $D_1 = C_2$, $D_{1h} = C_{2v}$, $D_{1d} = C_{2h}$.

| Normalizer: | Groups: |
|:---:|:---:|
| $K_h$ | $1$, $C_i$ |
| $D_{\infty h}$ | $C_n$, $S_{2n}$, $C_{nh}$ $\forall n \geq 2$; $C_s$ |
| $D_{(2n)h}$ | $C_{nv}$, $D_{nd}$, $D_{nh}$ $\forall n \geq 2$; $D_n$ $\forall n \geq 3$ |
| $I_h$ | $I$, $I_h$ |
| $O_h$ | $D_2$, $D_{2h}$, $T$, $T_d$, $T_h$, $O$, $O_h$ |

In the following subsections we do casework by normalizers. For each, subgroup with a given normalizer, we give a valid complement by name if it exists. In some normalizers, the name of a subgroup is not sufficient to identify it. This is because there are multiple copies of subgroups with that name in the normalizer. In such cases, we must identify which copy of the subgroup we care about. To do so, we give the normalizers in terms of a group presentation found with the help of GAP. Group presentations are essentially a set of generators and relations among the generators. We can then specify any specific subgroups of the normalizer using the generators of the normalizer.

### F.1  Normalizer: $K_h$

All the groups with this normalizer do have complements. Note that in Schönflies notation, $K_h$ is just the entire group $O(3)$. The only subgroups with $O(3)$ as normalizer are the trivial group $C_1$ and inversion $C_i$. Clearly for the trivial group the complement is $O(3)$. For inversion, the complement is just $SO(3)$.

Table 5: Groups with normalizer $K_h = O(3)$ and their complements.

| Group | Complement |
|:---:|:---:|
| $1$ | $K_h = O(3)$ |
| $C_i$ | $K = SO(3)$ |

### F.2  Normalizer: $D_{\infty h}$

Unfortunately, most of the groups in this case have no complements. We provide a proof of this fact here. We begin by showing no nontrivial cyclic group has a complement in $C_\infty$ (which is $SO(2)$ in Schönflies notation).

Table 6: Groups with normalizer $D_{\infty h}$ and their complements.

| Group | Complement |
|:---:|:---:|
| $C_s$ | $C_{\infty v}$, $D_\infty$ |
| $C_n$, $S_{2n}$, $C_{nh}$ $\forall n \geq 2$ | None |

**Lemma F.1.** *Let $C_n$ be a cyclic group of order $n \geq 2$ which is embedded in $C_\infty$. Note that it is a normal subgroup since all groups here are abelian. Then $C_n$ does not have a complement in $C_\infty$.*

*Proof.* **1** Suppose there was a complement $H$. By definition, for any $g \in C_\infty$ we have $g = ch$ for unique $h \in H$ and $c \in C_n$.

Next, note that $C_\infty$ is a divisible group. In particular, for any $g \in C_\infty$, there exists some $g'$ such that $g = (g')^n$. Let $g'$ be uniquely written as $h'c'$ for some $h' \in H$ and $c' \in C_n$. Then we have

$$g = (c'h')^n = (c')^n(h')^n = (h')^n$$

where we noted $(c')^n = e$. So $g$ is uniquely written as $g = ch$ for $c = e$ and $h = (h')^n$. But this holds for all $g$ so all elements of $C_\infty$ are just elements of $H$. This contradicts the fact that $H \cap C_n = \{e\}$. □

*Proof.* **2** For those familiar with exact sequences, one can consider the following alternative proof. Consider short exact sequence

$$1 \to C_n \to C_\infty \to C_\infty/C_n \to 1.$$

We can check that $C_\infty/C_n \cong C_\infty$. Existence of a complement for $C_n$ implies the above sequence is split, which by the splitting lemma Hatcher (2002) implies $C_\infty \cong C_\infty \oplus C_n$, a contradiction. □

We can now extend the lemma above to there being no complement of any cyclic group in $D_\infty$ (which is $O(2)$ in Schönflies notation).

**Lemma F.2.** *Let $C_n$ be a cyclic group of order $n \geq 2$ which is embedded in $D_\infty$ such that the rotation axis aligns with the infinite rotation axis in $D_\infty$. Note that it is a normal subgroup since all groups here are abelian. Then $C_n$ does not have a complement in $D_\infty$.*

*Proof.* Suppose there was a complement $H$. Consider $H' = H \cap C_\infty$. Clearly, we have $H' \cap C_n = \{e\}$. Next, for any $g \in C_\infty$, there is unique $c \in C_n$ and $h \in H$ such that $g = ch$. But $h = c^{-1}g \in C_\infty$ so $h \in H'$. So $H'$ is a complement of $C_n$ in $C_\infty$. But this contradicts Lemma F.1. □

Finally, we can prove that no subgroups except for $C_s$ in this case have complements. Note here that $K_h$ is just $O(3)$ and $K$ just $SO(3)$ in Schönflies notation.

**Theorem F.3.** *Consider any point group $A$ which has normalizer $D_{\infty h}$ in $K_h$. If $A$ has a nontrivial pure rotation, then it has no complement in $D_{\infty h}$.*

*Proof.* First, note that $K_h$ is the direct product of $K$ and inversion $C_i$. Suppose $A$ had a complement $H$.

We can split $A = A_e \sqcup A_i$ where $A_e = A \cap K$ is the subgroup of pure rotations and $A_i$ is a coset consisting of elements with an inversion. Similarly, we can split $H = H_e \sqcup H_i$ and $D_{\infty h} = D_\infty \sqcup D_i$ into subgroups of pure rotations and coset of elements with inversions.

We claim the elements of $G = A_e H_e$ form a group. Consider any $a, a' \in A_e$ and $h, h' \in H_e$. Since $A$ is a normal subgroup of $D_{\infty h} = AH$, we have $hA = Ah$ so $ha' = a''h$ for some $a'' \in A$. But since $h, a' \in K$, we have $a''h \in K$ so $a'' \in K$. Hence $a'' \in A_e$. Therefore,

$$(ah)(a'h') = a(ha')h' = a(a''h)h' = (aa'')(hh') \in A_e H_e.$$

So $G = A_e H_e$ is a group.

Next, since $H$ is a complement of $A$, clearly $A_e \cap H_e = \{e\}$. Since $G \leq AH = D_{\infty h}$, any $g = ah$ for unique $a \in A$ and $h \in H$ which by construction of $G$ are $a \in A_e$ and $h \in H_e$. So $H_e$ is certainly a complement of $A_e$ in $G$.

Now, we claim either $G = D_\infty$ or $G = C_\infty$. For any $g \in D_\infty$, since $H$ is a complement, there is a unique $a \in A$ and $h \in H$ such that $g = ah$. In particular, we note we must either have $a \in A_e$ and $h \in H_e$ or $a \in A_i$ and $h \in H_i$ to have the right inversion parity. One possibility is $G = A_e H_e = D_\infty$. For the other possibility, suppose $D_\infty - G$ is nonempty. Fix $g \in D_\infty - G$ and consider any $g' \in D_\infty - G$. Then $g = ah$ and $g' = a'h'$ where $a, a' \in A_i$ and $hh' \in H_i$. Now, note that $a^{-1}a'$ is the combination of 2 elements with odd parity in $i$ so $a^{-1}a' \in A_e$. Next, since $A$ is a normal subgroup, we have $h^{-1}A = Ah^{-1}$ and in particular,

$h^{-1}a^{-1}a' = a''h^{-1}$ for some $a'' \in A$. But since $h$ has odd parity and $a^{-1}a'$ has even parity in inversion, $a''$ must have even parity in inversion. Hence, we have

$$g^{-1}g' = (ah)^{-1}(a'h') = h^{-1}a^{-1}a'h' = a''(h^{-1}h').$$

Since $h, h'$ both have odd parity, $h'' = h^{-1}h'$ has even parity so in fact $g^{-1}g' = a''h''$ where $a'' \in A_e$ and $h'' \in H_e$. Therefore, $g^{-1}g' \in G$ so $D_\infty - G = gG$ is just a $G$-coset of $D_\infty$. We can similarly also show that $D_\infty - G = Gg$. Now, we claim $gG \cap C_\infty = \phi$. Suppose not. Then there is some $c \in gG \cap C_\infty$. Since $C_\infty$ is a divisible group, there is some $c'$ where $c = (c')^2$. But we must have $(c')^2 \in G$, a contradiction. Hence $gG \cap C_\infty = \phi$ so $G \cap C_\infty = C_\infty$. To conclude, we must have $g \in D_\infty - C_\infty$ and it is clear $gC_\infty$ would generate the remaining elements in $D_\infty$. So $G = C_\infty$ in this case.

From Table 4, we can see that $A_e$ must in fact be a nontrivial cyclic group. But then the above implies that $H_e$ is a complement of this cyclic group in either $G = D_\infty$ or $G = C_\infty$, which contradict Lemma F.2 and Lemma F.1 respectively. So $A$ cannot have a complement in $D_{\infty h}$. □

### F.3   Normalizer: $D_{(2n)h}$

All groups with this normalizer do have complements. We list the subgroup and its complement in Table 7. One presentation of $D_{(2n)h}$ is

$$\langle a, b, m | a^{2n}, b^2, m^2, (ab)^2, (am)^2, (bm)^2 \rangle.$$

Figure 14 depicts an example of a $D_{10h}$ object. The element $a$ correspond to a $2\pi/10$ rotation about the blue vertical axis, $b$ corresponds to a $\pi$ rotation about the red axis, and $m$ corresponds to a reflection across the mirror plane shown in orange.

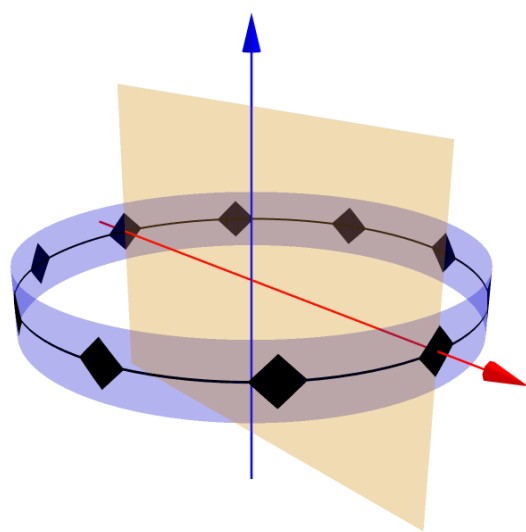

Figure 14: Object with symmetry $D_{10h}$. We can identify generator $a$ as the 10-fold rotation about the blue axis, generator $b$ as the 2-fold rotation about the red axis, and $m$ as the reflection over the plane shown in orange.

Table 7: Groups with normalizer $D_{(2n)h}$ and their complements.

| Group | Generators of group | Complement | Generators of a complement |
|-------|--------------------|------------|----------------------------|
| $C_{nv}$ | $a^2, m$ | $C_{2v}$ | $am, bm$ |
| $D_{nd}$ | $a^2, abm, m$ | $C_s$ | $bm$ |
| $D_{nh}$ | $a^2, b, m$ | $C_s$ | $am$ |
| $D_n$ | $a^2, b$ | $C_{2v}$ | $am, bm$ |

### F.4   Normalizer: $I_h$

This case is simple, we either have $I$ or $I_h$. Clearly we just need to add inversion to get a complement in the former case and in the latter case we can just take the trivial group.

Table 8: Groups with normalizer $I_h$ and their complements.

| Group | Complement |
|-------|------------|
| $I$ | $C_i$ |
| $I_h$ | $1$ |

### F.5   Normalizer: $O_h$

All subgroups in this case have complements as well. One presentation of $O_h$ is

$$\langle a, b, i | a^4, b^4, i^2, (aba)^2, (ab)^3, iaia^{-1}, ibib^{-1} \rangle.$$

Here, $a$ and $b$ are $\pi/2$ rotations about perpendicular axes and $i$ is just inversion.

Table 9: Groups with normalizer $O_h$ and their complements.

| Group | Generators of group | Complement | Generators of a complement |
|-------|--------------------|------------|----------------------------|
| $D_2$ | $a^2, b^2$ | $D_{3d}$ | $ab, ba^2, i$ |
| $D_{2h}$ | $a^2, b^2, i$ | $D_3$ | $ab, ba^2$ |
| $T$ | $ab, ba$ | $S_4$ | $a^2b, i$ |
| $T_d$ | $ab, ba, ai$ | $C_2$ | $a^2b$ |
| $T_h$ | $ab, ba, i$ | $C_2$ | $a^2b$ |
| $O$ | $a, b$ | $C_i$ | $i$ |
| $O_h$ | $a, b, i$ | $1$ | $\phi$ |

# G   Equivariant full SBS better than exact partial SBS

We provide an outline of the construction of the counterexample. It is easiest to explain this by introducing the concept of a wreath product on groups.

**Definition G.1** (Wreath product). Let $H$ be a group with a group action on some set $\Omega$. Let $A$ be another group. We can define a direct product group indexed by $\Omega$ as the set of sequences $(a_\omega)_{\omega \in \Omega}$ where $a_\omega \in A$. The action of $H$ on $\Omega$ induces a semidirect product by reindexing. In particular, for all $h \in H$ and sequences in $A^\Omega$ we define

$$h \cdot (a_\omega)_{\omega \in \Omega} = (a_{h^{-1}\omega})_{\omega \in \Omega}.$$

The resulting group is the unrestricted wreath product and denoted as $A \, \mathrm{Wr}_\Omega \, H$.

If rather than a direct product group $A^\Omega$, we restrict ourselves to a direct sum where all but finitely many elements in our sequence is not the identity, then we get the restricted wreath product denoted as $A \, \mathrm{wr}_\Omega \, H$.

Note that the direct sum and direct product are the same for finite $\Omega$ so the restricted and unrestricted wreath products also coincide in those cases.

Consider the space $\Omega = \{1, -1\}$ and an action of $D_4$ on $\Omega$ corresponding to the $A_2$ representation. Intuitively, if we think of $D_4$ as the rotational symmetries of a square in the $xy$-plane, this corresponds to how the $z$ coordinate transforms by flipping signs. Define a group $G'$ as $G' = C_2 \, \mathrm{wr}_\Omega \, D_4$. This is a group of order 32 and is `SmallGroup(32,28)` in the Small Groups library GAP. One presentation of this group is

$$\langle a, b, c | a^2, b^4, (ab)^4, c^2, bcb^{-1}c, (ac)^4 \rangle. \tag{1}$$

In this presentation, we can interpret $a, b$ as generators of $D_4$ and $c$ as the generator one copy of $C_2$.

Consider the group $G = G' \times G'$ defined using the direct product. We can write generators of $G$ as $a_1, b_1, c_1, a_2, b_2, c_2$ corresponding to two copies of those in the presentation given in equation 1 where generators with different indices commute. Define $S$ as the subgroup generated by $a_1, b_1^2, c_1, a_2, b_2^2, c_2$ and $K$ as the subgroup generated by $c_1 c_2$.

We can check that $N_G(S) = N_G(S, K) = G$. It is also not hard to check that $a_1 b_1, a_2 b_2$ generate a complement for $S$ in $G$. Hence, by Theorem 3.3, we know that an ideal $G$-equivariant SBS is possible for $S$. Hence, we know the size of the equivariant full SBS is $|S| = 256$.

Next, suppose we wanted a $G$-equivariant exact partial SBS. We can always generate this partial SBS by taking the orbit of some element $p$ under action by $N_G(S, K) = G$ where $\mathrm{Stab}_S(p) = K$. We claim we must also have $\mathrm{Stab}_G(p) = K$. Suppose not, then there must be some $g \in \mathrm{Stab}_G(p)$ such that $g \notin S$. However, we can check through casework or brute force that for all such $g$, either $gKg^{-1} \neq K$ or $g^2 \in S - K$. But would mean that there are elements not in $K$ which stabilize $p$ so $\mathrm{Stab}_S(p) \neq K$, a contradiction. Hence, no such $g$ can exist so $\mathrm{Stab}_G(p) = K$. Finally, by Orbit-Stabilizer theorem, this means the set must have size $|P| = |\mathrm{Orb}_G(p)| = |G|/|\mathrm{Stab}_G(p)| = (8^2 \cdot 2^4)/2 = 512$. This is larger than the equivariant full SBS.

In our supplementary material, we provide a script in GAP which verifies our claims above. In particular it performs a brute force check that $\mathrm{Stab}_S(p) = K$ implies $\mathrm{Stab}_G(p) = K$ for the group $G, S, K$ described above.

# H    Experiments

We provide some additional details on our experiments here. We also provide our code for running these experiments in the supplementary material.

## H.1    Triangular prism

### H.1.1    Obtaining an ideal SBS

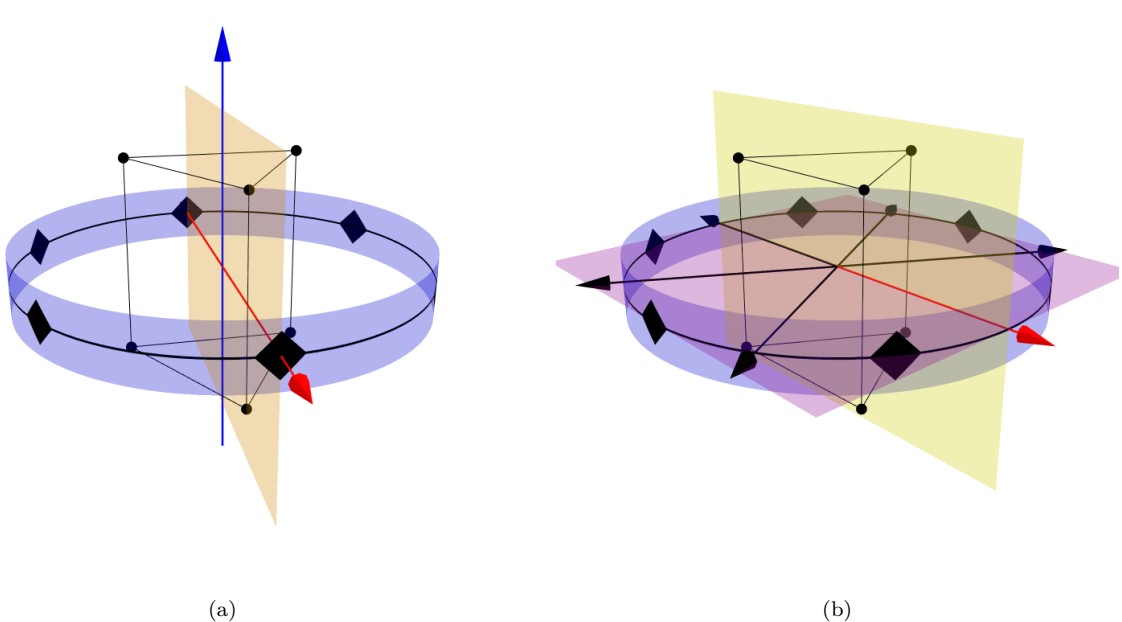

(a)(b)

Figure 15: (a) Triangular prism with $D_3$ symmetry and patterned cylinder with $D_{6h}$ symmetry. The generators are $a, b, m$ where $a$ is a $2\pi/6$ rotation about the blue axis, $b$ is a $\pi$ rotation about the red axis, and $m$ is a reflection across the orange plane. (b) An ideal symmetry breaking set for the triangular prism. A complement of $D_3$ in $D_{6h}$ is generated by the mirror planes shown here in yellow and purple. The vector in red is a symmetry breaking object with this complement as stabilizer. The orbit of this vector under the normalizer generates the other vectors shown in black.

Table 7 tells us $D_3$ is generated by $a^2, b$ and that a complement $H$ is generated by $am, bm$. By Theorem 3.3, we know that if we can pick some object $v$ with stabilizer $\mathrm{Stab}_{D_{6h}}(v) = H$, then the orbit of $v$ under $D_3$ gives an ideal equivariant SBS. The symmetry axis for $a$ and $b$ are depicted in Figure 15a and correspond to the $z$ and $x$ axes in the canonical orientation. The generators $am, bm$ of the complement are depicted as mirror planes in yellow and purple in Figure 15b.

In this case, we can see that the vector at the intersection of the mirror planes has the symmetries of the complement. This is depicted as the red vector in Figure 15b. One can further check it shares no symmetries with the triangular prism. The other 5 arrows in black are the other symmetry breaking objects we obtain by taking the orbit of the red arrow under action by $D_{6h}$. In the canonical orientation, this red vector normalized to unit length is

$$(\cos(\pi/6), \sin(\pi/6), 0) = (\sqrt{3}/2, 1/2, 0).$$

### H.1.2 Nonequivariant SBS

In addition to using an equivariant SBS as presented in the main paper, we also tried training with the non-equivariant SBS described in Section 3.2. Recall that the symmetry breaking objects here are a vector pointing to one of the vertices of the triangle projected in the $xy$ plane and a vector pointing up or down corresponding to which triangle we pick from. We fix one pair of vectors as our symmetry breaking object and train our equivariant model to match it with a vertex.

As shown in Figure 16a, our model is able to complete this. However, rotating the prism by 180° and feeding this rotated prism along with our symmetry breaking object, we find that our model outputs a vector which does not point to any vertex. This is shown in Figure 16b. Contrast this with the equivariant case in Figures 16c and 16d where our model still produces a vector which points to a vertex of the prism.

### H.1.3 Nonideal SBS

We can modify the nonequivariant SBS into an equivariant one by adding the additional objects needed for closure under the normalizer. Doing so we see there are 2 $S$-orbits in this nonideal equivariant SBS shown in Figure 17. This corresponds to a degeneracy of 2 as defined in Section H.1.3.

From the discussion in Section 3.4, we expect this SBS to be less efficient than an ideal one. We first train using only a symmetry breaking object from one of the $S$-orbits. We see that when given objects from that orbit, the network correctly outputs vectors pointing to vertices of the prism but fails when given objects from the other $S$-orbit. However, if we use objects from both orbits in training, the output vectors point to vertices when given objects from either $S$-orbit. Hence, we need to use at minimum 2 symmetry breaking objects during training to guarantee correct behavior for all objects in the SBS compared to only needing one example to train for an ideal SBS.

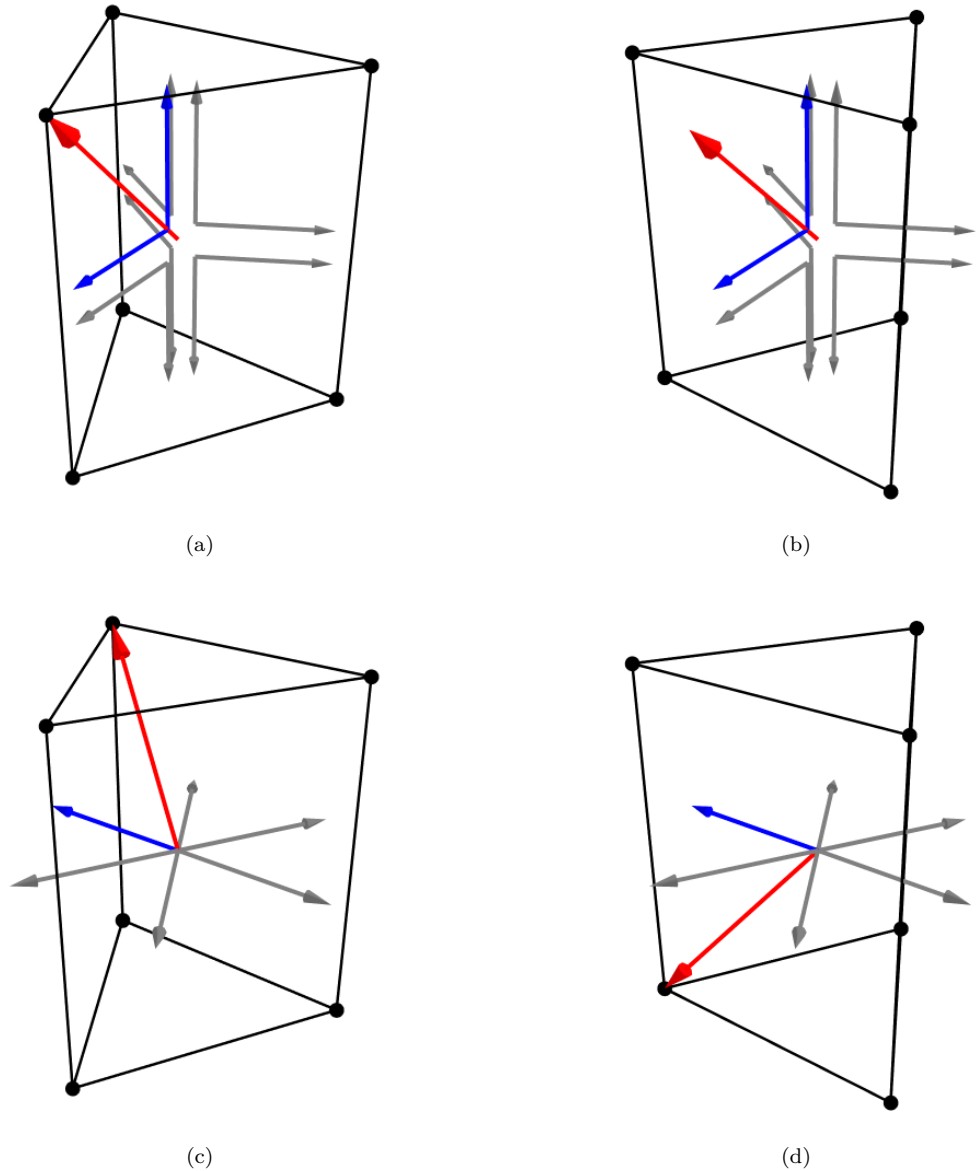

(a)

(b)

(c)

(d)

Figure 16: (a) Output (red) generated by our model and symmetry breaking object (blue) given that is chosen from a non-equivariant SBS (b) Output (red) generated by our model and symmetry breaking object (blue) given when our prism is rotated by 180° (c) Output (red) generated by our model and symmetry breaking object (blue) given that is chosen from an equivariant SBS (d) Output (red) generated by our model and symmetry breaking object (blue) given when our prism is rotated by 180°

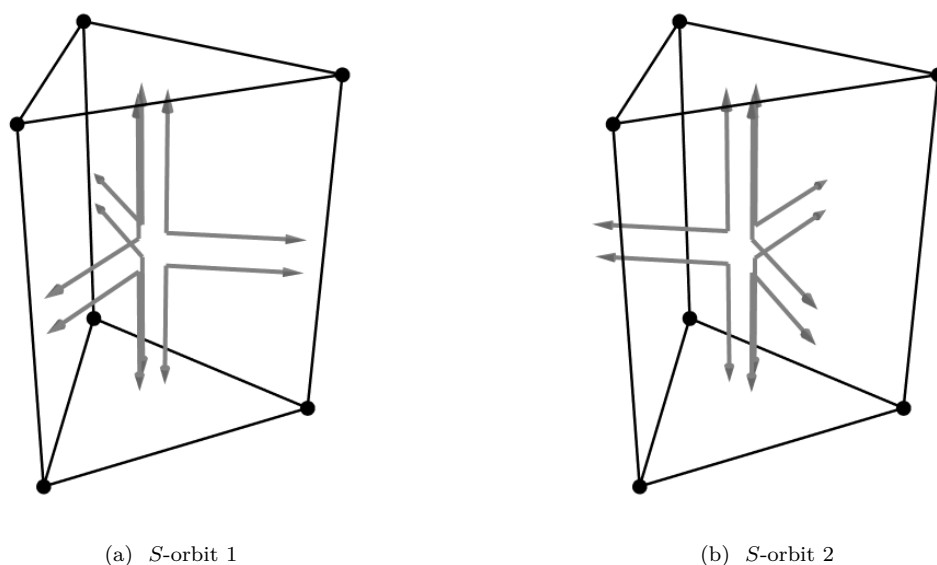

(a) *S*-orbit 1           (b) *S*-orbit 2

Figure 17: (a) *S*-orbit corresponding to the original nonequivariant SBS (b) Additional *S*-orbit needed which makes the set closed under the normalizer.

Table 10: Results of training using a nonideal equivariant SBS. If we only see one of the $S$-orbits in training, the network fails on the unseen orbits. If we see both $S$-orbits then the network behaves correctly.

| Seen in training | | Outputs | |
|---|---|---|---|
| $S$-orbit 1 | $S$-orbit 2 | $S$-orbit 1 | $S$-orbit 2 |
| Yes | No | | |
| No | Yes | | |
| Yes | Yes | | |

## H.2 Octagon to rectangle

### H.2.1 Obtaining an ideal partial SBS

In this scenario, we want $G$-equivariance for $G = O(3)$ and we have $S = D_8$ and partial symmetry $K = D_2$.

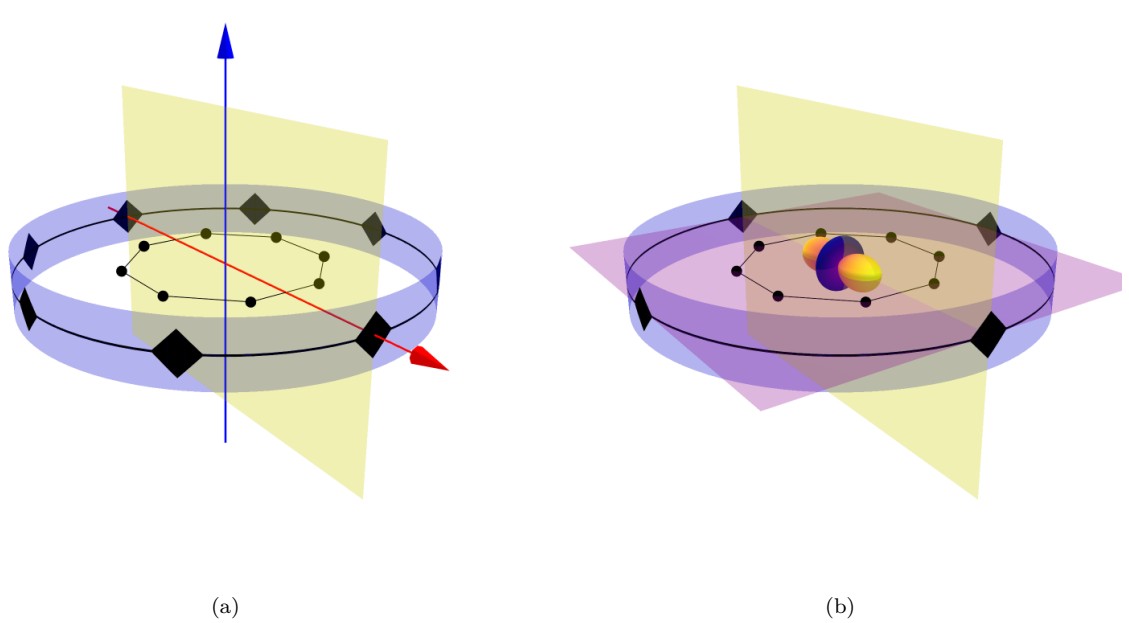

(a)                       (b)

Figure 18: (a) Octagon with $D_8$ symmetry we input and patterned cylinder with $N_G(S, K) = D_{8h}$ symmetry. The generators are $a^2, b, m$ where $a^2$ is a $2\pi/8$ rotation about the blue axis, $b$ is a $\pi$ rotation about the red axis, and $m$ is a reflection across the orange plane. (b) A symmetry breaking object which generates an ideal partial SBS. Note that in addition to the $D_2$ symmetry, this object is also symmetric under reflections across the purple and yellow planes.

We follow Algorithm 3 to understand what symmetry a canonical symmetry breaking object needs to generate an ideal partial SBS. First, we note $S = (e, D_8)$ and $K = (e, D_2)$. The first few steps are straightforward and we can evaluate them directly.

| Operation | Evaluated |
|---|---|
| $N_1 \leftarrow \texttt{Normalizers}[(\mathbf{S})]$ | $\texttt{Normalizers}[D_8] = (e, D_{16h})$ |
| $(n, (\mathbf{N_2}))) \leftarrow \texttt{Normalizers}[(\mathbf{K})]$ | $\texttt{Normalizers}[D_2] = (e, D_{4h})$ |
| $N_2 \leftarrow (g_S^{-1} g_K n, (\mathbf{N_2}))$ | $(e, D_{4h})$ |
| $N \leftarrow N_1 \cap N_2$ | $(e, D_{16h}) \cap (e, D_{4h}) = (e, D_{4h})$ |
| $N' \leftarrow (e, (\mathbf{S})) \cap \mathbf{N_2}$ | $(e, D_8) \cap (e, D_{4h}) = (e, D_4)$ |

The next step is

$$(Q_1, \phi) \leftarrow \texttt{Quotient}[N, (g_S^{-1} g_K, (\mathbf{K}))]$$

where we need to create a quotient group. Here we note a presentation of $N = (e, D_{4h})$ is

$$\langle a, b, m | a^4, b^2, m^2, (ab)^2, (am)^2, (bm)^2 \rangle.$$

The normal subgroup $(g_S^{-1} g_K, (\mathbf{K})) = (\mathbf{e}, \mathbf{D_2})$ then consists of $e, a^2, b, a^2 b$. Hence we can set the cosets

$$X = \{a, a^3, ab, a^3 b\} \qquad Y = \{m, a^2 m, bm, a^2 bm\}$$

and we can check these generate the quotient group and we have relations $X^2 = Y^2 = (XY)^2 = e$. Hence we have

$$Q_1 = \langle X, Y | X^2, Y^2, (XY)^2 \rangle.$$

Finally, setting

$$\phi(a) = \phi(a^3) = \phi(ab) = \phi(a^3b) = X$$
$$\phi(m) = \phi(a^2m) = \phi(bm) = \phi(a^2bm) = Y$$

defines our homomorphism $\phi$.

The next step is

$$Q_2 \leftarrow \phi(N').$$

We note $N' = (e, D_4)$ which is generated by $a, b$. We can find that $\phi(a) = X$ and $\phi(b) = \phi((a^3)(ab)) = \phi(a^3)\phi(ab) = X^2 = e$. Hence,

$$Q_2 = \langle X | X^2 \rangle.$$

The next step is

$$C \leftarrow \texttt{FindComplement}[Q_1, Q_2]$$

which in this case one can check to be generated by $Y$. This gives

$$C = \langle Y | Y^2 \rangle.$$

Since $C$ exists, the next step is

$$(h, (\mathbf{H})) \leftarrow \phi^{-1}(\mathbf{C}).$$

The elements of $C$ are $e, Y$ and from the definition of $\phi$, we can see that $\phi^{-1}(C)$ consists of the elements

$$\phi^{-1}(e) = \{e, a^2, b, a^2b\} \qquad \phi^{-1}(Y) = \{m, a^2m, bm, a^2bm\}.$$

We can check that the group consisting of these elements is in fact generated by $a^2, b, m$ which is $(e, D_{2h})$. Hence

$$(h, (\mathbf{H})) = (\mathbf{e}, \mathbf{D_{2h}}).$$

If an object has this symmetry then it can generate an idea partial SBS. In this case a simple choice is a $l = 2$ object of even parity. This is depicted in Figure 18b. If we align it to the $x$ axis, this would correspond to $(0, 0, 1, 0, 0)$ using real spherical harmonic conventions. Applying Algorithm 2 we would obtain the set of $l = 2$ objects "parallel" to an edge of the octagon.

### H.3   BaTiO$_3$ experiment

#### H.3.1   Atom matching algorithm

We would like to predict atom distortions. However, our data consists of atom coordinates in the initial and target structures not necessarily in the same order. Hence, we need an algorithm to match similar atoms together so we know how much they are distorted. This process is complicated by the fact that we have periodic boundary conditions with periodicity determined by lattice vectors and that our atoms may be translated within the lattice. We assume our structures are given in the same rotational orientation.

For our algorithm, we use the insight that the distorted atom should still have similar vectors to neighboring atoms. Hence, we can compute a signature for an atom $a$ by taking the difference of the positions of that atom and all other atoms in the lattice. We call the set of position differences for atom $a$ from all other atoms the signature $\sigma_a$ of $a$. Note in our implementation, we also separate out the atom types in addition to the position differences.

Next, we need a way to compare signatures. Suppose we had atom $a$ from the initial structure and atom $a'$ from the target structure. Certainly, if they are different atom types, we assign a cost of $\infty$ to this pairing.

Otherwise, we look at their signatures. However, we actually need to optimally pair the other atoms to do so. For a pair of atoms $b$ and $b'$ from the initial and target structures, we can give a cost of $\infty$ if they are different atoms and $\sigma_a[b] - \sigma_{a'}[b']$ otherwise. If the atoms are similar, this should be small, but it may also be shifted by lattice parameters from the smallest it could be. So we just look at all small lattice shifts $L$ and set the smallest $||\sigma_a[b] - \sigma_{a'}[b'] + L||^2$ as the cost of pairing $b, b'$. This gives a cost matrix $M_{b,b'}$. With all the costs, we can run the matching algorithm in Crouse (2016) to match the atoms. The cost of the assignment is the difference in the signatures of $a, a'$.

Finally, we can match the atoms using the comparison of signatures. We create a cost matrix where $C_{a,a'}$ is the difference in signatures of $a, a'$ from the initial and target structures. We then run another iteration of the algorithm from Crouse (2016) to find our matching.

Because we only ever use differences of atoms, this matching algorithm is independent of translations.

### H.3.2 Translation invariant loss

In the previous experiments, we could simply use a MSE loss of the vector differences. However, for crystals we wanted to have a loss which is invariant under translation. We realize that our matching algorithm in fact produces such a loss. By storing the matching information, we can effectively compute the same loss without having to run the matching algorithm every time. This is what we use to evaluate our model.

### H.3.3 Symmetrically related outputs

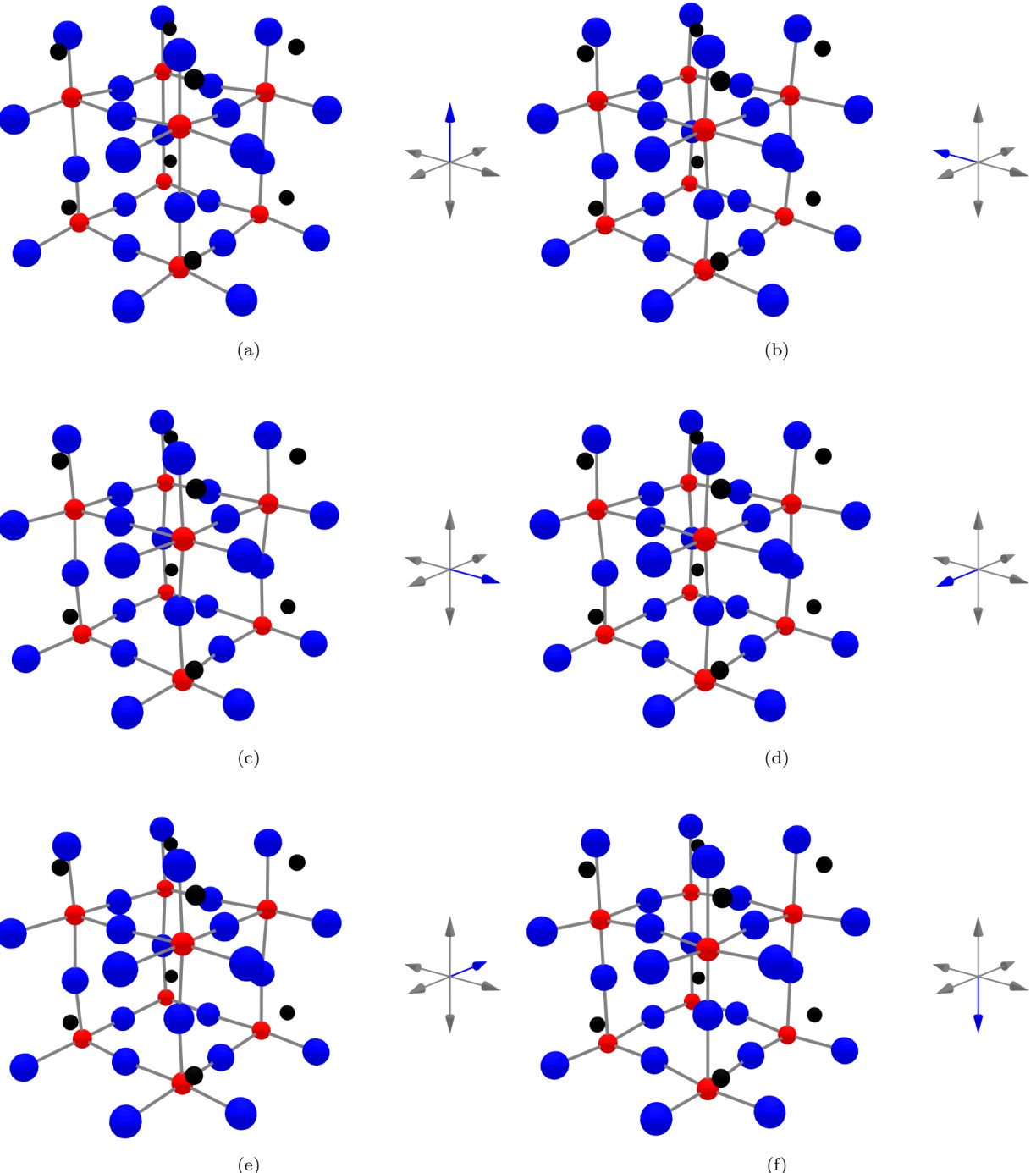

Figure 19: Distortions of a highly symmetric crystal structure of $BaTiO_3$ when provided with each of the possible symmetry breaking objects in our ideal equivariant partial SBS.

Table 11: Values of various quantities which help distinguish the high symmetry and low symmetry structures. Our models here try to distort the high symmetry structure to the low symmetry one.

| Structure | SB object | Bond length average | Bond length variance | Ti-O1-Ti | Ti-O2-Ti | Ti-O3-Ti |
|---|---|---|---|---|---|---|
| High symmetry | | 2 | 0 | 180° | 180° | 180° |
| Low symmetry | | 2.003417 | 0.01392 | 180° | 171.80° | 171.80° |
| Model | None | 2 | 0 | 180° | 180° | 180° |
| Model | $(1, 0, 0)$ | 2.003417 | 0.01392 | 180° | 171.80° | 171.80° |
| Model | $(-1, 0, 0)$ | 2.003417 | 0.01392 | 180° | 171.80° | 171.80° |
| Model | $(0, 1, 0)$ | 2.003417 | 0.01392 | 171.80° | 180° | 171.80° |
| Model | $(0, -1, 0)$ | 2.003417 | 0.01392 | 171.80° | 180° | 171.80° |
| Model | $(0, 0, 1)$ | 2.003417 | 0.01392 | 171.80° | 171.80° | 180° |
| Model | $(0, 0, -1)$ | 2.003417 | 0.01392 | 171.80° | 171.80° | 180° |

