# OpenReview forum: "Equivariant Symmetry Breaking Sets"
_TMLR — Accepted by TMLR_

### Review · Reviewer_A4Dc · 2024-08-12

**Summary Of Contributions:**

This paper proposes a method to address the problem of symmetry breaking in equivariant neural networks. The goal is to make it so that an equivariant neural network can have outputs of lower symmetry than its input. This is done by sampling elements from a set that depends on the symmetry of the input. The authors characterize these sets that can break the symmetry, the goal being to make them as small as possible using group theoretical notions. They also look at partial symmetry breaking, for which the goal is to obtain outputs that break some of the symmetry but not all. The authors also perform some toy experiments to show how the method works.

**Audience:**

Yes

**Broader Impact Concerns:**

This work doesn't pose specific concerns, but some discussion of impact areas (e.g. chemistry) could be considered.

**Claims And Evidence:**

Yes

**Requested Changes:**

**Major changes**
- I think definition 3.2 is confusing. The set B which is defined as equivariant instead of sigma. Only functions can be equivariant not outputs of functions. I understand that this is motivated by the fact that if B satisfies definition 3.2, then an equivariant sigma exists, but I think this should be mentioned explicitly.
- "Intuitively, we expect a smaller SBS to be better and this is indeed true." The last part of the statement is too strong. I did not find the experiments convincing in supporting this statement (see below). This should be changed to "we intuitively expect this to be true because..." or something like that.
- I don't understand the motivation behind the definition 4.1 of the partial (non-exact) SBS. I thought that the motivation for partial SBSs is that we might know that the output has some symmetry? Likewise, I don't understand the sentence, "In general, we can always break more symmetry than needed and still obtain our desired output.". If the desired output has a symmetry, we shouldn't be allowed to break it? Overall, I think that the necessity of this definition should be questioned it would be beneficial to the paper to remove it. If it can't be removed, it should be justified much more clearly.
- I found that the experimental methods described in section 6 do not tie in clearly with the method presented in section 5. The method introduced seemed quite systematic, but then some unjustified choices are made in the experiments. It seems like I am reading about a completely different method in section 6. It is important to show if and how the method follows the algorithm introduced in section 5.
- "in Appendix H.1.3 we show that a nonideal SBS is less efficient than an ideal one." I think this statement is misleading. A notion of efficiency was introduced earlier in the paper, according to which an ideal SBS is by definition more efficient than an ideal one. If that is the notion this statement refers to, then it doesn't add anything. However, it seems like efficiency here is taken in the sense of better empirical performance. I don't find the evidence presented in Appendix H.13 against the nonideal SBS conclusive. If I understand correctly, the SBS are built to be nonideal (2 orbits) but in one version of the experiment the network is trained only on one orbit? That doesn't make sense to me, the network should be trained on the full SBS. Indeed when it is, it produces good results. I think the discussion is misleading, it seems like the non-ideal SBS is fine. Possibly a more challenging task would be necessary to see if nonideal SBSs are really worst.
- One thing that is not clear is how the element of the symmetry breaking set should be in general be given as input to the equivariant neural network. This is worth elaborating in the methods section.

**Minor changes**

- Lemma 2.1:
    - If the group action is transitive, there is only one orbit so it should not be necessary to restrict the domain to the orbit
    - Why is the version 2.1 in the main text an "only if" and the version in the appendix an "if and only if"?
- "However, in some cases we truly have a symmetry breaking sample even if there is no noise.": I don't understand this statement, spontaneous symmetry breaking requires noise, else what mechanism chooses how the symmetry is broken?
- "The intuition is that we want to maximize the symmetries of the symmetry breaking objects but only in the directions “orthogonal" to S." I don't find the intuition obvious. If I understand correctly, given S we want to find the complement of S in N_G(S), K. If this complement exists, then by making the elements of the SBS symmetric with respect to K, but asymmetric with respect to S, we obtain an ideal SBS. If this is correct, I suggest fleshing out this intuition more, as it can help the reader understand, but the current version is not clear enough, especially what is meant by orthogonal.
- I think the method you introduce for construction an Equivariant SBS in section 5.4 can be seen as a case of equivariance by canonicalization (Kaba, 2023) since the SBS is constructed only for the canonical subgroup S, and the SBS for any input is obtained by applying the canonicalizer to this SBS. I think this connection is worth mentioning.
- "Since we want this to be a full-symmetry breaking task, we require chirality in our prism." This seems quite ad hoc, I don't understand why this should be done. Isn't it sufficient that the group acts freely on the symmetry breaking objects to break all the symmetry as explained in the theory section?

**Strengths And Weaknesses:**

**Strengths**

- The paper addresses an interesting and as far as I understand, potentially important problem of equivariant deep learning.
- As far as I could check, the theoretical results are correct. The presented framework is self-contained.
- The discussion of the limitations of the method is detailed and helpful.

**Weaknesses**

- I found that sometimes the explications of the paper are slightly unclear or imprecise. Some of the claims also require more justification. See Requested Changes.
- The experiments are a bit lackluster as they do not showcase any real application. But this is acceptable for a mostly theoretical paper.

---

> ### Author Response · Authors · 2024-09-05
> **Response to Reviewer A4Dc**
>
> We thank the reviewer for their extremely thorough reading of our work and detailed feedback.
>
> Regarding the requested changes
> ### Major changes
> * This is a very good point! We have updated the paper to explicitly mention this before giving Definition 3.2.
> * Fair enough, we removed “and this is true.”
> * These are great questions! By “we can always break more symmetry,” what we mean is that a model not constrained to be as symmetric can always relearn the broken symmetry through training. For example, nonequivariant architectures can still learn to be equivariant through data augmentation. The reason we did not directly define partial SBS (Definition 4.1) as exact partial SBS (Definition 4.2) is discussed in Section 4.5 with a contrived counterexample in Appendix G.
> * Thank you for the feedback, we reorganized the experimental section and hope it makes the connection more clear
> * Thanks for feedback. We agree that from an empirical performance perspective, the task is too simple to show anything. The intention was to give a tangible demonstration of our definition of degeneracy.
> * We did not specify this as we intended flexibility for how a practitioner may want to incorporate the symmetry breaking sets. However, we added some examples of how one may do so.
>
> ### Minor changes
> * Lemma 2.1
>     - Good catch! That is leftover from a previous version where we wanted to get rid of the transitive action condition
>     - Fair point, I guess if the wording was “we can choose an equivariant function” then it is an only if statement (but still if only if for the appendix), but if the wording is “Let $f:X\to Y$ be an equivariant function” then it should if and only if. The difference being that in the first case it could be possible no equivariant $f:X\to Y$ exists but in the latter case it is implied that an equivariant $f:X\to Y$ exists. We believe the first wording is clearer and have changed the definition accordingly.
> * What we meant is that in some cases, the actual output is actually as symmetric as the input but is observed as less symmetric only because of a noisy measurement. However, in explicit and spontaneous symmetry breaking, even with a perfect measurement we find a less symmetric output. This is different from how the system evolved to the less symmetric output which, as you point out, is due to noise in the system and the symmetric state being unstable in spontaneous symmetry breaking.
> * Yes, your understanding is correct. We updated the intuition and placed it after the theorem. Hopefully it is clearer.
> * Yes absolutely! We added a reference.
> * Indeed it is a bit ad hoc. We wanted to reuse the same prism example from section 3.2, however without chirality it would actually be a partial symmetry breaking task. If we use O(3) equivariance instead of SO(3), then the vector to each point has a mirror symmetry in the $D_{3h}$ group.
>
> Once again, thank you for taking the time to give a careful review of our work. We hope our answers above have addressed your concerns. Please let us know if you have any other feedback.

---

> > ### Comment · Reviewer_A4Dc · 2024-09-12
> >
> > I am satisfied the authors response to my requests and recommend acceptance of this paper.

---

### Review · Reviewer_PoBz · 2024-08-13

**Summary Of Contributions:**

The paper Equivariant Symmetry Breaking Sets provides a mathematical analysis of symmetry breaking in machine learning architectures. It distinghuishes explicit symmetry breaking, in which the underlying phenomenon is asymmetric, and spontaneous symmetry breaking, in which underlying phenomenon being modelled is symmetric, but the observed data is not due to noise. There has been a recent surge of interest in relaxed equivariances and symmetry breaking in the community. This paper provides a formal treatment of different forms of symmetry breaking, and ways to incorporate symmetry breaking in neural network architectures through symmetry breaking sets (SBS). Lastly, some experiments are provided to demonstrate these techniques in practice.

**Audience:**

Yes

**Broader Impact Concerns:**

No ethical implications.

**Claims And Evidence:**

No

**Requested Changes:**

Overall, I think this is an interesting work that will definitely be of interest to members in the community. In particular, I value the formal group theoretical analysis of spontaneous symmetry breaking and results. I think this is the strongest part of the paper.

I have two main concerns relating this work. The first is that it is not clear how architectures are being used, which makes it very hard or impossible or hard to assess how proposed symmetry breaking will actually be able to solve the formulated goals. The second concern is about the experimental validation, which lacks details to sufficiently assess the made claims.

## Critical points

> Make explicit how architectures are used.

The paper focuses on group theoretical analysis of the architecture. However, the stated goals, claims and experimental validation all imply that the architectures are somehow used to solve particular tasks. The paper does not mention or formalize how these architectures are used, which makes it hard to follow and verify some of the statements. The paper does mention that there is a "distribution of data" and imply some empirical fit on the data "equally likely". However, without specifying how this fit is made and what task is being performed it is hard to judge what properties (in terms of symmetry breaking) are desired in the architecture. I think this modelling context should be made explicit when claims are made about certain (symmetry breaking) properties being desirable in certain contexts.

> Concerns about experimental validation

Although I believe the method could be very useful, I don't find the experimental validation sufficient enough to support the made claims. This is in part because I found the description of the experiments to be very vague, which made it hard to assess and interpret the results. The authors do mention that the experiments are more of a proof of concept. However, I would suggest either clearly describing the experiments or leaving out the experimental valdiation all together. Currently, it is not clear what the training data is, what objectives are being optimized, and how results are evaluated.


## Minor points


> Difference between explicit / spontaneous symmetry breaking well explained, but Figure 1 very confusing

The destinction between explicit and spontaneous symmmetry breaking is a central subject of the paper. I find that the difference is quite well explained in the text, in whether the underlying governing laws are symmetric. Although this was clear, and further formalized in Sec. 2, I found Figure 1 really confusing and not helpful in its current form. It starts to mentions magnetic moments and external fields without ever introducing these. Also, "B" is never introduced.

> Relation to other works on relaxed equivariance

The authors mentions some architectures that encorporate relaxed equivariant architectures (Huang et al., 2023; van der Ouderaa et al., 2022; Wang et al., 2023; 2022b). Can the authors elaborate what is meant by "If all symmetrically related lower symmetry
outputs are equally likely in the data, then relaxed networks will see the distribution as symmetric and fail
to break symmetry ". Could authors elaborate on this and explain why this would not actually be desirable in such cases? From a generalization perspective, wouldn't it be desirable to have a symmetric architecture in such case? This comes back to the previous question that it is not clear how architectures relate to the task at hand or predictions being made.

**Strengths And Weaknesses:**

### Strengths.

There has been recent interest in the community to symmetry breaking in neural network architectures. The paper recognizes an important destinction between different kinds of symmetry breaking in physics, namely asymmetries due to asymmetries in the underlying governing laws (explicit symmetry breaking) and asymmetry in observations while the underlying process is symmetry (spontaneous symmetry breaking). The paper provides a group theoretical treatment of different types of symmetry breaking and how such symmetry breaking can be incorporated in the architecture. This seems to be the main strength of the paper. The paper also provides some experiments to demonstrate the approach in practice.

### Weaknesses.

Although the reviewer appreciates the generality of stated results and group theoretical formalization of the proposed concepts, the effective methodological contribution from a machine learning perspective is very minimal. The method seems to boil down to adding additional inputs to the equivariant layers. In terms of method, there seems quite some overlap with Smidt et al. (2021). I do not find this too worrying, given there is value in a generalizing theory connecting this approach to other methods and puts it in a broad group theoretical framework. The experimental validation on the other hand is very hard to follow and lacks the details to really assess claims being made.

> Analysis is limited to architecture, but claims suggest architectures are used in some training scheme.

What confuses me about this work is that most of the analysis seems to be limited to the architecture. However, some of the claims imply that these architectures are used for predictions without ever mentioning how these architectures are used or trained and how predictions are made. If claims are made about using the architecture in practice, it should be made explicit how these architectures are trained or used to make predictions.

---

> ### Author Response · Authors · 2024-09-05
> **Response to Reviewer PoBz**
>
> We thank you for your reading of our work and detailed feedback. We appreciate that the reviewer values our general group theoretical treatment of the problem.
>
> From our understanding, much of the criticism in the review seems related to the type of tasks we are considering. In particular, we suspect the description of spontaneous symmetry breaking, how it translates to a machine learning task, and the difficulty it presents for equivariant models may have been confusing to the reviewer. We hope that this response and updated paper helps clarify the setting we consider.
>
> In this response we first address the minor points as they help clarify the spontaneous symmetry breaking setting. We then return to the critical points.
>
> ## Minor points
> > Difference between explicit / spontaneous symmetry breaking well explained, but Figure 1 very confusing
>
> We appreciate the feedback on Figure 1. We realize the physics of magnetism may be unfamiliar in the machine learning community and may be less appropriate for this audience. We have expanded the description of ferromagnetism and moved it as Section 2.1.2. In addition, we added the double well potential as an example as Section 2.1.1. We hope the examples are clearer.
>
> > Can the authors elaborate what is meant by "If all symmetrically related lower symmetry outputs are equally likely in the data, then relaxed networks will see the distribution as symmetric and fail to break symmetry ". Could authors elaborate on this and explain why this would not actually be desirable in such cases? From a generalization perspective, wouldn't it be desirable to have a symmetric architecture in such case?
>
> These are great questions and really highlight how explicit and spontaneous symmetry breaking tasks differ. In both cases, we would like to predict a lower symmetry output given a higher symmetry input. However, we emphasize that the key difference is in spontaneous symmetry breaking we have a **set** of symmetrically related **equally correct** lower symmetry outputs while in explicit symmetry breaking it is possible to have only one correct output. In both cases, using an equivariant model to predict a/the lower symmetry output is impossible if we have a symmetric architecture (result of Lemma 2.1 or equation (5) in Smidt et. al. 2021).
>
> Relaxed networks can learn to be less symmetric if the distribution of the training data is less symmetric. This is fine for explicit symmetry breaking since such the data distribution would not be symmetric due to some external unknown factors. However if the distribution is symmetric then the relaxed network does not learn to break symmetry and by Lemma 2.1 would fail to give a low symmetry output. Not only would they not generalize, they fail to behave correctly for the training!
>
> Your intuition of a symmetric architecture being desirable is really one of the main motivations of this work. This is why we introduce the set-valued functions in Section 2 and we define what it means for a function to remain symmetric in a spontaneous symmetry breaking sense with Definition 2.4. The goal of the rest of the paper is to present and analyze a relatively natural way to construct the spontaneous symmetry breaking (SSB) function described in Definition 2.4. Returning to the critical points we can say the following.
>
> (continued below)

---

> > ### Author Response · Authors · 2024-09-05
> > **continuation (critical points)**
> >
> > ## Critical points
> >
> > > Make explicit how architectures are used
> > > What confuses me about this work is that most of the analysis seems to be limited to the architecture
> >
> > Before addressing this point, we want to ensure we have the same definition of architecture and framework. To us, an architecture refers to a specific type of neural network model while a framework refers to a strategy for designing types of models. We view our method as a framework for constructing a SSB function given we know how to construct equivariant functions. Hence it is applicable for **any** equivariant architecture (ie. equivariant CNN, equivariant transformer, equivariant GNN, etc.).
> >
> > If you mean limited to the framework, we want to emphasize the idea of introducing an additional symmetry breaking input is a very natural one. The analysis follows from:
> > * avoiding the consequences of Lemma 2.1 (introduce additional symmetry breaking input)
> > * satisfying equivariance of Definition 2.4 (introduce symmetry breaking set, normalizer, etc.)
> > * optimizing (minimizing size of SBSs, main theorems + corollaries, classification)
> > All of this is needed to fully understand the framework and is analyzed in full generality. It does not depend on any specific training scheme or task.
> >
> > We hope this clarifies our work and would appreciate it if the reviewer can point out any specific claims about the general framework they found unclear or insufficiently supported.
> >
> > > Concerns about experimental validation
> >
> > We appreciate the feedback about the experiments. These are really meant as an example of how one can apply our framework to an existing equivariant architecture (equivariant message-passing GNN built with e3nn). The tasks are all extremely simple spontaneous symmetry breaking toy examples which are impossible for existing equivariant architectures to solve. We have edited Section 6 and hope the new version helps clarify things.
> >
> > We hope our responses have addressed your concerns. Please let us know if you have any additional feedback.

---

> > > ### Comment · Reviewer_PoBz · 2024-09-21
> > >
> > > The authors have addressed the key concerns raised, and their revisions have clarified several points of confusion. The promised improvements appear to enhance the quality of the work. I recommend the paper for acceptance.

---

### Review · Reviewer_PWxe · 2024-08-30

**Summary Of Contributions:**

Motivated from physical problems where the outputs have less symmetries than the inputs, this paper proposed a symmetry breaking framework for equivariant neural networks (ENNs). The key idea is to feed the symmetry breaking parameters as additional input to the ENNs. Such symmetry breaking parameters are obtained from a symmetry breaking set (SBS) constructed from the input and output symmetries. The authors provided characterizations of equivariant SBS under different settings: (1) full symmetry breaking (ideal versus non-ideal case); (2) partial symmetry breaking (ideal versus non-ideal). The authors also presented algorithms in constructing equivariant SBS for (1) and (2). This framework is illustrated via numerical experiments on synthetic examples and crystal structure deformation.

**Audience:**

Yes

**Broader Impact Concerns:**

No ethical concerns.

**Claims And Evidence:**

Yes

**Requested Changes:**

1. Conceptual and empirical comparison between this approach and the approach in Smidt et al.(2021)

2. Discussion or clarification of the "efficiency" claim.

3. Most of the discussions and the numerical examples seem to concern with one input-output pair. How well can this framework be applied to a large dataset with input-output pairs having potentially different symmetries? This needs to be further elaborated.

4. The current formulation relies on the input and output having different amounts of symmetries. Can this be alternatively characterized as the same group with different actions in the input and output space? It will be nice to add some discussions on this front.

**Strengths And Weaknesses:**

Strengths

1. The paper is clearly written and explains well the motivation of spontaneous symmetry breaking

2. The formulation of equivariant symmetry breaking sets (SBS) is interesting. The connection of constructing equivariant SBS to finding complements of normal subgroups is novel.

Weaknesses

1. Equivariant SBS proposed in this work is closely related to the learnable symmetry breaking approach in Smidt et al.(2021). However, a clear justification of why equivariant SBS is more desirable (in what settings) is lacking, and a empirical comparison of these two approaches can significantly strengthen the paper.

2. The authors mentioned their framework allows "efficiently" symmetry breaking. However, it is not clear to me how efficient this framework is. The various discussions of efficiency (e.g. Sec 3.4, Sec 4.4) seem to only concern the characterization of non-ideal SBS. Given such characterizations, what can we infer about efficiency in terms of computational complexity? Or do the authors consider some other notion of efficiency?

---

> ### Author Response · Authors · 2024-09-05
> **Response to Reviewer PWxe**
>
> We thank you for your reading of our work and detailed feedback. We appreciate that the reviewer finds the formulation of symmetry breaking sets (SBS) interesting, the motivation of spontaneous symmetry breaking clearly explained, and paper clearly written.
>
> We would like to address the weakness and requested changes.
>
> ## Weaknesses/requested changes
> * >Comparison with Smidt et. al. 2021
>
>     While motivated by the same underlying problem (Lemma 2.1 in our paper  and equation (5) in Smidt et al. 2021), the goals are quite different. As a result it does not make sense to make an empirical comparison. The main similarity is the addition of a symmetry breaking input, which by itself is a very natural idea. In Smidt et. al., the goal is to learn this additional input which helps identify what additional input is needed to identify the observed lower symmetry output. This is great for explicit symmetry breaking and is related to relaxed equivariant frameworks. However, it provides little insight for spontaneous symmetry breaking problems. We have expanded Section 2 which hopefully makes the distinction clearer.
> * > notion of efficiency
>
>     This is a good question. Our notion of efficiency can be thought of as a variant of sampling efficiency and directly corresponds with the intuition of minimizing the size of a SBS. How it would translate to computational efficiency depends on many design factors which are beyond the scope of this work.
> * > input-output pairs having potentially different symmetries
>
>     Good question, most of the analysis is understanding how to use only input (and output) symmetries to create and sample from a SBS. As long as we have a consistent way of inputting symmetry breaking objects, we in principle use the algorithms from Section 5 to pick different SBSs for different inputs and insert the corresponding objects into the equivariant model. The restriction to only single types of input is just to simplify the examples. We mention this in the revised paper. If the reviewer feels strongly, we can try adding an example with multiple types of input symmetries.
> * >Can this be alternatively characterized as the same group with different actions
>
>     Trying to use different group actions is far too broad. The fundamental issue really is the difference in input and output symmetries as described in Lemma 2.1. Perhaps the new example in Section 2.1.1 helps in clarifying this.
>
> We thank you again for your time in reviewing our work. Please let us know if you have any other concerns.

---

> > ### Comment · Reviewer_PWxe · 2024-09-25
> >
> > I thank the authors for the detailed response, and the updated manuscript. I find the new double-well example in Sec 2.1 motivating and clear. I appreciate the updated Section 6 that carefully explains the task and architecture, with nice illustrations.
> >
> > Regarding the comparison with Smidt et al. (2021): Based on the updated manuscript, I agree with the authors that the goals are different. However, both methods intend to find a symmetry-breaking object (symmetry-breaking set in this work, and symmetry-breaking parameters in Smidt et al.). Can the authors comment on if there are any explicit connections between them (e.g will the learnable parameters in Smidt et al. learn the SBS in this work, if set-up properly)? Perhaps there are fundamental distinctions between these two approaches that I am missing. In any case, a more detailed discussion would be great.
> >
> > Overall, I find the updated version satisfying, and thus recommend acceptance of the paper.

---

### Author Response · Authors · 2024-09-05
**Main paper revisions**

We thank all the reviewers for their time and helpful comments. We have incorporated much of the feedback in a revised version of our paper and would like to highlight the major changes here.

## Intro/symmetry breaking problem
Based on the reviews, we believe our explanation of the symmetry breaking problem can be improved. We have added the classic double well potential as an example and explicitly step through the definitions of Section 2. In addition, we moved the ferromagnetism example to Section 2 and added some more background about magnetism.

## Reorganization of experiments section
We have reorganized the experiments section to explicitly point out the inputs, outputs, and their respective symmetries. This hopefully better aligns with the procedure described in Section 5.

---

### Comment · Action_Editor_DPZ6 · 2024-09-18
**Reminder for reviewers**

Hi reviewers,

Thank you all for your work! The authors have now posted their revised manuscript and responses to your comments. When you have a chance, please interact with the authors and then, when satisfied, make an official recommendation.

If you have any questions, feel free to reach out to me.

AE

---

### Author Response · Authors · 2024-10-25
**Thank you**

We would like to thank all the wonderful reviewers and the action editor for their time,  insightful comments, and positive feedback. This has been extremely helpful for improving our paper. We have uploaded the camera ready version of our work and link to the code. To address the requested minor revision, we added a brief section (Section 6) discussing how our method relates to Smidt et al. (2021) and some other existing works.

---

### Decision · Action_Editor_DPZ6 · 2024-09-27

**Recommendation:** Accept with minor revision

**Comment:**

One of the reviewers noted that one comment of their's remained unanswered by the authors after the rebuttal. Namely,

"The following question is not addressed by the authors yet. I would appreciate if some discussions can be added to the camera-ready version. “Regarding the comparison with Smidt et al. (2021): Based on the updated manuscript, I agree with the authors that the goals are different. However, both methods intend to find a symmetry-breaking object (symmetry-breaking set in this work, and symmetry-breaking parameters in Smidt et al.). Can the authors comment on if there are any explicit connections between them (e.g will the learnable parameters in Smidt et al. learn the SBS in this work, if set-up properly)? Perhaps there are fundamental distinctions between these two approaches that I am missing. In any case, a more detailed discussion would be great.” "

I think the authors can and should address this in the camera ready version, hence the need for minor revision.

Otherwise, all reviewers seemed satisfied with the state of the revised manuscript and the unanimous recommendation of acceptance led to my decision to accept.

**Audience:**

All reviewers noted that the work was appropriate for publication in TMLR. The problem of incorporating/learning symmetries from real world data is an important one of general interest in the machine learning community, and this paper's exploration of symmetry breaking extends ideas in this area to new research directions. Therefore, I believe this paper will be of interest to the TMLR community.

**Claims And Evidence:**

2 of 3 reviewers initially said they believed the claims and evidence were well supported. The reviewer who did not, noted that the experimental results were limited and did not support some of the claims the authors made about the utility of their method. The authors agreed that the experiments were simple (the point of them being that way was to explain the theory and demonstrate how the ideas they discuss might be made practical), and the toned down the language in the revised manuscript to allow for more accurate claims. The reviewer said that this was satisfactory.

All 3 reviewers noted the theoretical results were novel and solid, and given that this was a major contribution of the paper, I believe the evidence in the submission is accurate, convincing and clear.